EMBO
Molecular Medicine

# Combination therapies induce cancer cell death through the integrated stress response and disturbed pyrimidine metabolism

Goetz Hartleben[1,2,3], Kenji Schorpp[4], Yun Kwon[1,2,3], Barbara Betz[1,2,3] (iD), Foivos-Filippos Tsokanos[1,2,3], Zahra Dantes[5], Arlett Schäfer[5], Ina Rothenaigner[4], José Manuel Monroy Kuhn[6], Pauline Morigny[1,2,3], Lisa Mehr[1,2,3], Sean Lin[4] (iD), Susanne Seitz[1,2,3], Janina Tokarz[7], Anna Artati[7], Jerzy Adamsky[3,7,8,9], Oliver Plettenburg[10,11], Dominik Lutter[6], Martin Irmler[12], Johannes Beckers[3,12,13], Maximilian Reichert[5,14,15], Kamyar Hadian[4], Anja Zeigerer[1,2,3], Stephan Herzig[1,2,3,16,*] (iD) & Mauricio Berriel Diaz[1,2,3,**] (iD)

## Abstract

By accentuating drug efficacy and impeding resistance mechanisms, combinatorial, multi-agent therapies have emerged as key approaches in the treatment of complex diseases, most notably cancer. Using high-throughput drug screens, we uncovered distinct metabolic vulnerabilities and thereby identified drug combinations synergistically causing a starvation-like lethal catabolic response in tumor cells from different cancer entities. Domperidone, a dopamine receptor antagonist, as well as several tricyclic antidepressants (TCAs), including imipramine, induced cancer cell death in combination with the mitochondrial uncoupler niclosamide ethanolamine (NEN) through activation of the integrated stress response pathway and the catabolic CLEAR network. Using transcriptome and metabolome analyses, we characterized a combinatorial response, mainly driven by the transcription factors CHOP and TFE3, which resulted in cell death through enhanced pyrimidine catabolism as well as reduced pyrimidine synthesis. Remarkably, the drug combinations sensitized human organoid cultures to the standard-of-care chemotherapy paclitaxel. Thus, our combinatorial approach could be clinically implemented into established treatment regimen, which would be further facilitated by the advantages of drug repurposing.

**Keywords** cancer metabolism; integrated stress response; metabolic vulnerabilities; pyrimidine metabolism; tricyclic antidepressants
**Subject Categories** Cancer; Pharmacology & Drug Discovery; Signal Transduction

## Introduction

Cancers differ in their genetic driver mutations, which infers a necessity to identify the right drug fitting to the corresponding mutation. Thus, complex diagnostics need to be in place to determine biomarkers for therapy success. Furthermore, intra-tumoral heterogeneity adds another layer of complexity, resulting in insufficient molecular

---

1   Institute for Diabetes and Cancer, Helmholtz Center Munich, Neuherberg, Germany
2   Joint Heidelberg-IDC Translational Diabetes Program, Heidelberg University Hospital, Heidelberg, Germany
3   German Center for Diabetes Research (DZD), Neuherberg, Germany
4   Assay Development and Screening Platform, Institute of Molecular Toxicology and Pharmacology, Helmholtz Zentrum München, Neuherberg, Germany
5   Klinik und Poliklinik für Innere Medizin II, Klinikum rechts der Isar, Technical University of Munich, Munich, Germany
6   Institute for Diabetes and Obesity, Neuherberg, Germany
7   Research Unit Molecular Endocrinology and Metabolism, Helmholtz Center Munich, Neuherberg, Germany
8   Lehrstuhl für Experimentelle Genetik, Technische Universität München, Freising-Weihenstephan, Germany
9   Department of Biochemistry, Yong Loo Lin School of Medicine, National University of Singapore, Singapore City, Singapore
10  Institute of Medicinal Chemistry, Helmholtz Zentrum München, German Research Center for Environmental Health (GmbH), Neuherberg, Germany
11  Institute of Organic Chemistry, Center of Biomolecular Drug Research (BMWZ), Leibniz University Hannover, Hannover, Germany
12  Institute of Experimental Genetics, Helmholtz Zentrum München GmbH, Neuherberg, Germany
13  Technische Universität München, Chair of Experimental Genetics, Freising, Germany
14  Center for Protein Assemblies (CPA), Technische Universität München, Garching, Germany
15  German Cancer Consortium (DKTK), Munich, Germany
16  Chair Molecular Metabolic Control, Technical University Munich, Munich, Germany
    *Corresponding author. Tel: +49 89 3187 1046; E-mail: stephan.herzig@helmholtz-muenchen.de
    **Corresponding author. Tel: +49 89 3187 1093; E-mail: mauricio.berrieldiaz@helmholtz-muenchen.de

targeting of the whole tumor cell population. Therefore, targeting cancer vulnerabilities, which could be largely independent from the genetic driver, represents a promising alternative therapeutic approach. Cancer cells are characterized by a high nutrient demand in order to sustain bioenergetics and biosynthesis for survival and proliferation (Pavlova & Thompson, 2016). If deprived of nutrients, cancer cells rewire their metabolism for compensation, leading to the ability to survive a transient nutrient shortage. Such a state of metabolic stress leaves the cell sensitive to further perturbations that in turn represent attractive targets for therapeutic intervention. In this context, one promising treatment concept is reflected in pharmacologically induced cellular starvation. Hence, drugs that target pathways important for tumor metabolism could mimic nutrient starvation, thereby triggering secondary vulnerabilities similarly as under conditions of actual nutrient limitations. Especially mitochondrial targeting drugs were shown to induce metabolic stress and chemo-sensitivity in cancer cells (Lee *et al*, 2018), in agreement with the finding that oncogenic induction of OXPHOS can lead to chemotherapy resistance via ROS formation (Lee *et al*, 2017).

Cancer cells rely on mitochondria for both the synthesis of building blocks during proliferation and bioenergetics, and pharmacological or genetic interference with mitochondrial function is known to attenuate the tumorigenic potential. For example, depletion of cancer cells of mitochondrial DNA impaired their potential to form tumors in vivo, demonstrating the requirement of mitochondrial function for tumorigenesis (Tan *et al*, 2015). This was further shown by inhibition of oxidative phosphorylation (OXPHOS) to slow glioblastoma progression in vivo with the small molecule Gboxin, which was identified in a high-throughput viability screen, representing a more therapy-related example (Shi *et al*, 2019). One distinct class of mitochondrial targeting drugs are mitochondrial uncouplers. These compounds uncouple the electron transport chain from mitochondrial ATP production. While the toxicity of first-generation mitochondrial uncouplers precluded them from systemic application, second-generation uncouplers demonstrate an excellent safety profile and are used in preclinical and clinical studies to treat obesity and other metabolic diseases (Tao *et al*, 2014). The mitochondrial uncoupler niclosamide and its ethanolamine salt form niclosamide ethanolamine (hereafter referred to as NEN) have been shown to possess anti-tumor efficacy in a variety of preclinical tumor rodent models (Alasadi *et al*, 2018). However, used as a single drug, niclosamide has only minor therapeutic efficacy at its maximal tolerated dose. Thus, combinatorial treatments in which the single drugs synergize to induce cancer cell death could accentuate drug efficacy and thereby provide novel approaches in cancer therapy.

Using phenotypic high-throughput drug screenings, we identified the dopamine receptor antagonist domperidone as well as several structurally related tricyclic antidepressants (TCAs) synergizing with the mitochondrial uncoupler NEN to induce cancer cell death through activation of two interrelated metabolic stress response pathways, the integrated stress response (ISR) and the catabolic CLEAR network. Induction of these pathways mainly driven by the transcription factors CHOP and TFE3, respectively, led to the induction of the pyrimidine-degradative enzyme UPP1 and the downregulation of the pyrimidine biosynthesis machinery. As a consequence, enhanced degradation and decreased synthesis of pyrimidines resulted in lethal DNA damage. Of clinical importance, we demonstrated that the identified drug combinations sensitized tumor-

derived organoids from pancreatic cancer patients to the standard-of-care chemotherapy paclitaxel. Translation of proposed combinatorial treatments into clinical application is facilitated by the advantages of drug repurposing, as well as the option to implement it into established treatment regimen, potentially overcoming drug-resistance and reducing side effects.

# Results

## Mitochondrial uncoupling induces targetable metabolic vulnerabilities in cancer cells

We aimed to interfere pharmacologically with mitochondrial function at a non-toxic dose to force the cell to rewire signaling and metabolism and adapt to this low-level stress, thus imposing increased vulnerability onto the cancer cell. One known cellular response to mitochondrial OXPHOS intervention is the upregulation of glycolytic flux, in order to maintain cellular ATP levels. As a proof-of-principle, we tested if the compensatory glycolysis was required for survival under mitochondrial uncoupling conditions. Uncoupling using NEN and FCCP (carbonyl cyanide 4-(trifluoromethoxy)phenylhydrazone) in the colorectal cancer cell line HCT116, the glioblastoma U87 and the pancreatic adenocarcinoma line BxPC3 did not cause significant toxicity. However, concomitant inhibition of glycolysis using 2-deoxyglucose (2-DG), a molecule exerting glycolysis inhibitory effects, but not 2-DG alone caused significant cell death in all 3 cell lines tested (Figs 1A and EV1A). Interestingly, this was accompanied by a significant drop in ATP levels (Fig EV1B). Of note, this drop in ATP did not fully explain the cell death, since already 2-DG alone led to a similar decrease in ATP without inducing cell death. We obtained similar results under low serum cell culture conditions, which combined with low-dose NEN treatment caused significant cell death (Fig EV1C). This showed that mitochondrial uncoupling provoked a metabolic rewiring, which compromised the ability of cancer cells to cope with additional metabolic stress.

In order to identify drugs that synergize with mitochondrial uncouplers to induce cancer cell death, we performed a high-throughput phenotypic viability screen using a library of 1,280 FDA-approved drugs (Prestwick collection). Hits were defined as compounds that induced cell death in NEN co-treated cells but not in vehicle-treated cells and were categorized in several biological functions, with G protein-coupled receptor (GPCR)-targeting drugs being overrepresented (Fig 1B), including dopamine receptor antagonists and beta-blockers (Fig 1B). Therefore, we performed a second screen using a focused GPCR library comprising 680 compounds (Fig 1C) and integrated the two screens. The dopamine receptor antagonist domperidone scored as hit in both screens, indicating that it represented a promising candidate drug. Interestingly, tricyclic antidepressants were represented with several compounds in both screens including trimipramine, desipramine, and clomipramine. The structural similarity between the TCAs suggest a high degree of robustness of the observed synergistic effects in inducing cell death. Taken together, we identified several GPCR-targeting drugs that synergize with mitochondrial uncoupling to induce cancer cell death in a novel combinatorial approach.

Noteworthy, phenotypic screens cannot account for polypharmacology, which implies that a specific phenotypic readout can be

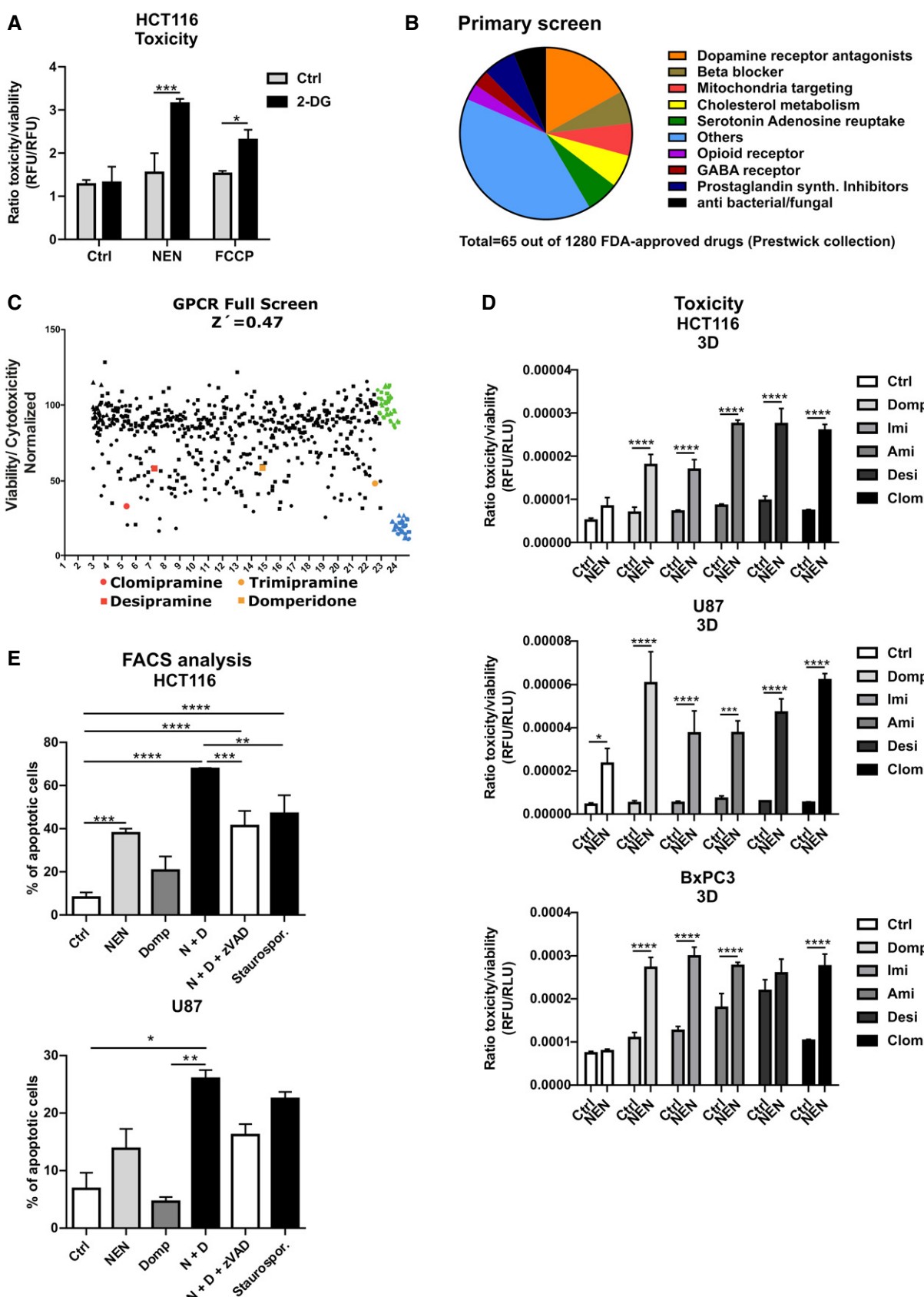

**Figure 1.**

**Figure 1.  Mitochondrial uncoupling induces targetable metabolic vulnerabilities.**

A   HCT116 cells were treated as indicated (Control (Ctrl), 2-DG 100 mM; NEN 1.2 μM; FCCP 2.5 μM). Graph shows the ratio of toxicity and viability measured after a 24-h treatment.

B   Pie chart depicting numerical proportions of indicated drug categories on total hits from the phenotypic screening using the FDA-approved drug library.

C   Plot shows results of GPCR screen as ratio of cell toxicity and viability. Domperidone and TCAs highlighted in red and orange, vehicle controls in green, and positive toxicity controls in blue. Z′ factor was calculated according to the formula $Z' = 1 - (3(\theta p + \theta n)/(\mu p - \mu n))$, where $p$ is the positive control, $n$ is the negative control, $\theta$ is the standard deviation, and $\mu$ is the mean. For hit selection, a threshold of lower than 3 standard deviations from the median of the negative/vehicle population was set.

D   Ratio of toxicity and viability of HCT116, U87 and BxPC3 cells grown in spheroids (3D) and treated as indicated for 24 h (BxPC3) or 48 h (HCT116, U87) (NEN 1.2 μM; domperidone (Domp), imipramine (Imi), desipramine (Desi), and amitriptyline (Ami), each 30 μM; clomipramine (Clomi) 20 μM).

E   Percentage of apoptotic cells, determined by measuring Annexin V- and PI-positive HCT116 or U87 cells by FACS. The effects were assessed after a 24-h or 48-h treatment, respectively (NEN (N) 1.2 μM; domperidone (Domp, D) 30 μM, zVAD-FMK (zVAD) 100 μM, Staurosporine 100 μM). Cells were pre-treated for 1 h with the zVAD-FMK inhibitor prior to adding the drugs.

Data information: Data in (A, D and E) are presented as mean (SD) ($N = 3$) and were analyzed by two-way ANOVA with Tukey *post hoc* test (A, D) or one-way ANOVA with Tukey *post hoc* test (for E, upper graph) HCT116 cells) or by Kruskal–Wallis with Dunn's *post hoc* tests (for E, lower graph) U87 cells). Significance is indicated for multiple comparisons as indicated. *$P < 0.05$; **$P < 0.01$; ***$P < 0.001$; ****$P < 0.0001$. Exact $P$-values for all comparisons are listed in Appendix Table S1.

achieved through different routes. Therefore, the overall diversity in the molecular structures of the hits reflected either that different drug targets merged on the same effector pathways, or a certain degree of polypharmacology of hit compounds resulted in overlapping targets accounting for the observed effect. Therefore, we decided to include domperidone, as one of the top hits in both screens, as well as selected TCAs in subsequent analyses for drug validation and identification of a potential common mechanism of action (MOA). In addition to the TCA hits desipramine and clomipramine, we added imipramine to the validation pipeline instead of its analogue trimipramine, since it is characterized by higher water solubility and demonstrated efficacy against glioblastoma and small-cell lung cancer (SCLC) in preclinical mouse models (Jahchan *et al*, 2013; Shchors *et al*, 2015). In addition, desipramine, which was a hit in the screen, represents the active molecule of imipramine. Clomipramine is another derivative of imipramine, containing one additional Cl-residue. Although amitriptyline was not among the hits, we chose to include it for further analyses in order to obtain more structure–activity relationship (SAR) information, as well as due to its excellent safety profile and high water solubility, which facilitates preclinical application. Upon initial determination of dose–response curves for individual drug treatments in HCT116 cells (Appendix Fig S1), we validated the synergistic cell death induction upon dual treatment with NEN and the screening hits in HCT116, U87, and BxPC3 cells. Strikingly, treatment with all drug combinations, but not with the single drugs, induced cell death in 3D and 2D cell cultures, as assessed by fluorescence and luminescence-based assays for cytotoxicity as well as caspase 3/7 activation, respectively (Figs 1D and EV1D). We confirmed induced cell death upon dual drug treatments with NEN using flow cytometry-based apoptosis assays (Figs 1E and EV1E). The effect of combined NEN + Domp treatment was at least partly rescued by concomitant treatment with the pan-caspase inhibitor zVAD-FMK (Fig 1E). In contrast, when we treated primary hepatocytes with the individual drugs as well as the respective combinations with NEN, we did not observe any considerable induction of cell death, as assessed by caspase 3/7 activation and in contrast to staurosporine treatment as positive control (Fig EV1F). This was in line with our hypothesis that untransformed cells are less sensitive to the identified drug combinations, suggesting favorable characteristics of respective potential therapeutic approaches with respect to adverse side effects.

## Drug combinations synergistically induce the integrated stress response (ISR)

To elucidate the mechanism of cell death induction, we performed transcriptome analysis from cells treated either with vehicle, domperidone, NEN, or in combination. Gene ontology (GO) analysis of biological processes of combinatorially treated versus control cells revealed differentially regulated metabolic and stress pathway signatures (Appendix Fig S2). Interestingly, "response to starvation" was one of the upregulated gene sets, whereas "mitochondrial gene expression" was downregulated, which was in line with our initial idea, to induce a starvation-like response with our pharmacological treatment. In line with a halt in proliferation, we found "ribosome biogenesis" and "DNA replication" being downregulated. Further prominently upregulated signatures fell in the categories "response to ER stress" and "response to unfolded proteins". The genes of these categories belong to a conserved gene expression program subsuming the responses to a variety of cellular insults, namely the integrated stress response (ISR). The ISR includes components responsive to metabolic and mitochondrial stress, which can exert pro-survival or pro-apoptotic effects depending on the extent and the duration of the stress trigger. The heat map in Appendix Fig S3 depicts differentially expressed genes, showing elevated mRNA levels upon individual treatments, which were markedly stronger upregulated under combined treatment conditions. Notably, this indicated drug interaction in upregulating the expression of specific genes might have been responsible for the synergism in apoptosis activation. These genes also included the ISR transcription factors DDIT3 (CHOP), a major driver of apoptosis and DDIT4 (ATF4). QPCR analysis of genes involved in the ISR including CHOP, ATF4, and the CHOP target GADD34 confirmed the transcriptomics results. Single treatment with NEN, domperidone, or TCAs led to modest increase in ISR marker gene expression; however, double treatment synergistically induced the upregulation of ATF4, CHOP, and GADD34 (Figs 2A and EV2A). Furthermore, we confirmed elevated protein levels of the ISR transcription factors CHOP and ATF4 as well as induced phosphorylation/activation of the upstream regulator eIF2alpha upon individual and, markedly more pronounced, double drug treatments, including quantifications of the respective immunoblot analyses in the 3 different cell lines (Figs 2B and EV2B and C, Appendix Fig S4A and B).

The ISR has a dual cellular role, contributing to survival under transient stress and inducing apoptosis if stress is persistent. In order to characterize the role of the ISR in the cellular response to the drugs, we individually knocked down the two integral members ATF4 and CHOP and treated the cells with a combination of NEN and domperidone. Strikingly, CHOP knockdown, but not ATF4 knockdown, prevented the increase in apoptosis upon combinatorial drug exposure (Figs 2C and EV2D). Similarly, pharmacological intervention in the ISR using the inhibitor ISRIB (Sidrauski et al, 2013) prevented the induction of CHOP and of its target gene Gadd34, but did not decrease ATF4 expression significantly (Fig 2D). Congruently, ISRIB treatment also attenuated cell death induction upon treatment with NEN in combination with domperidone or TCAs in different cancer cell lines in 2D as well as spheroid cultures (Figs 2E and F, and EV2E and F). Taken together, we showed that the induction of CHOP mediated the apoptotic response to the identified drug combinations indicating that under the given conditions, the ISR was not cell protective but represented a central component of the cell death-inducing mechanism.

## Combinatorial treatments induce the Coordinated Lysosomal Expression and Regulation (CLEAR) network associated with a blockage in autophagy

To further understand the underlying mechanism of drug-induced cell death, we analyzed the transcriptome data applying categorization of regulated genes using WikiPathways (https://www.wikipathways.org). Similar to the GO terms, we identified several metabolic and stress response pathways regulated upon single and combinatorial treatment, including mitochondrial gene expression and OXPHOS, cholesterol metabolism, DNA damage and pyrimidine metabolism as well as autophagy in cancer (Appendix Fig S5). Autophagy is a cellular response to nutrient-limiting conditions, in which cellular components are degraded in the lysosomal autophagosome network to provide energy substrates. The transcription factors TFE3 and MITF are master regulators of autophagy and regulate a network of genes required for cellular degradation machinery, the so-called CLEAR network (Palmieri et al, 2011). Under conditions of metabolic stress, TFE3 and MITF induce a shift to catabolic metabolism by upregulation of the CLEAR genes, in order to secure cell survival. Recently, it was shown that lung

cancer initiation was dependent on AMPK-induced TFE3 nuclear translocation and lysosome biogenesis (Eichner et al, 2019). Importantly, the CLEAR network and the ISR crosstalk in the regulation of downstream target genes, and TFE3 was even described to be part of the ISR in the response to ER stress (Martina et al, 2016). Interestingly, we identified TFE3 and MITF to be induced upon combinatorial drug treatments, with a markedly more pronounced response of TFE3 expression (Figs 3A and EV3A). In agreement with these results, we also observed increased expression of selected TFE3 and MITF target genes, which was particularly strong upon combined treatments (Figs 3A and EV3B, Appendix Fig S6). In line with CLEAR network activation, we also measured enhanced nuclear protein levels of TFE3 and MITF upon combinatorial drug treatment, indicating nuclear translocation and thus activation of these transcription factors (Fig EV3B). In congruence with the role of TFE3 and MITF in lysosome biogenesis and autophagy, the levels of the late endosomal/lysosomal marker Lamp1 were markedly increased under combined treatment conditions, determined by immunofluorescence (IF) staining in HCT116 cells (Fig 3B). CLEAR network activation is a well-established sign for the activation of autophagy. Thus, we measured the autophagy marker LC3-II and the autophagy receptor p62 by immunofluorescence staining and immunoblot analysis in HCT116 cells (Fig 3B and C). Interestingly, LC3-II was induced already upon single but significantly more pronounced upon combined treatment with NEN and domperidone, indicating enhanced formation of autophagosomes. Surprisingly, p62 showed a similar pattern, suggesting additional p62 accumulation with double treatment. This was also observed using the TCA drugs (NEN + Ami and NEN + Imi), confirming the overlapping response of the treatments with the different drug combinations (Fig EV3C and D). This regulation pattern suggested that in addition to induced autophagosome formation, shown by the elevated LC3-II levels, the accumulation of p62 indicating a blockage of autophagosome clearance. This was in agreement with elevated Lamp1 levels, demonstrating reduced lysosomal autophagosome clearance, despite activation of autophagy, thus resulting in an overall reduction in autophagic flux. To directly test this, we analyzed autophagic flux by treating HCT116 cells under control, NEN, and NEN + Domp conditions with 20 μM chloroquine (CQ) for 3 h and determined the ratio of LC3-II levels between CQ-treated and CQ-untreated cells under the different drug treatment conditions (Klionsky et al, 2016).

---

**Figure 2. Drug combinations synergistically induce the integrated stress response.**

A  Relative ATF4 and CHOP mRNA expression levels in HCT116 cells upon the indicated treatments determined by qPCR (NEN (N) 1.2 μM; domperidone (Domp, D), imipramine (Imi), desipramine (Desi), and amitriptyline (Ami), each 30 μM; clomipramine (Clomi) 20 μM).

B  Immunoblot analysis of CHOP, ATF4, phosphorylated (p-) and t-) eIF2alpha (eIF2a) of HCT116 cells treated as indicated for 16 h (NEN (N) 1.2 μM; domperidone (Domp), amitriptyline (Ami), each 30 μM).

C  Caspase activity in HCT116 cells either transfected with control (Ctrl), ATF4- or CHOP-targeting siRNAs and treated as indicated (NEN (N) 1.2 μM, domperidone (D) 30 μM).

D  Relative mRNA expression levels of the indicated genes upon drug treatments (Control (Ctrl), NEN (N) 1.2 μM, domperidone (D) 30 μM) and inhibition of the ISR using ISRIB (1 μM), determined by qPCR.

E  Caspase activity in HCT116 cells grown as monolayer (2D) and treated as indicated for 24 h (Control (Ctrl), NEN 1.2 μM, domperidone (D) 30 μM, ISRIB 1 μM).

F  Ratio of toxicity and viability in HCT116 spheroids treated as indicated for 48 h (NEN 1.2 μM; domperidone (Domp), imipramine (Imi), desipramine (Desi), and amitriptyline (Ami), each 30 μM; clomipramine (Clomi) 20 μM; ISRIB 1 μM).

Data information: Data are presented as mean (SD) and were analyzed by one-way ANOVA with Tukey *post hoc* test (A, upper left and lower left graphs, (N = 3), (D (N = 3)), Kruskal–Wallis with Dunn's *post hoc* tests (A, upper right and lower right graphs, (N = 5, besides NEN + Clomi (N = 4) and NEN + Desi (N = 4)) or two-way ANOVA with Tukey *post hoc* test (C (N = 3), E (N = 4), F (N = 4). In (A), right graphs, significance is indicated for the comparison of combinatorial treatments (NEN + TCAs) to controls. *P < 0.05; **P < 0.01; ***P < 0.001; ****P < 0.0001. Exact P-values for all comparisons are listed in Appendix Table S1.

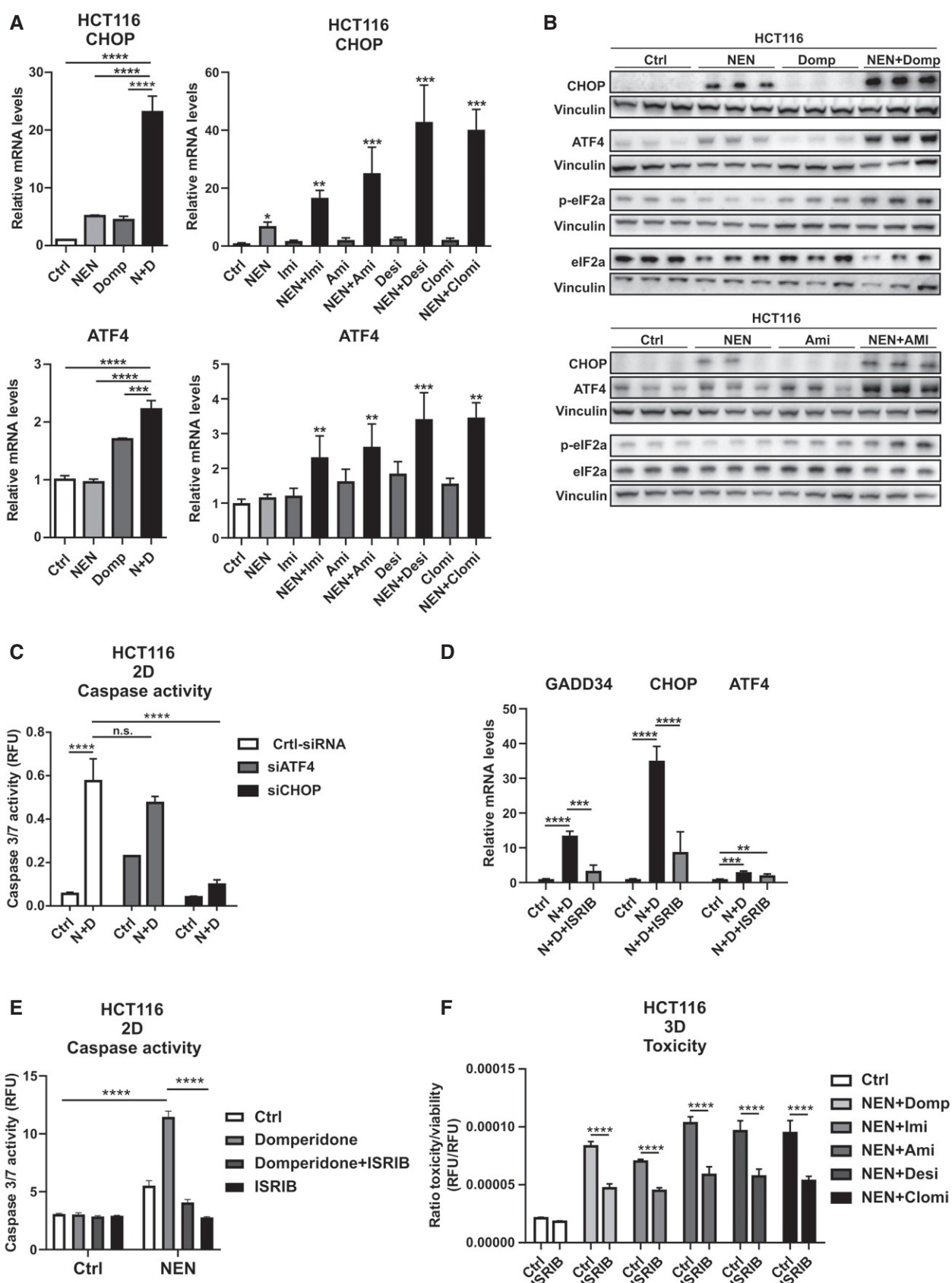

Figure 2.

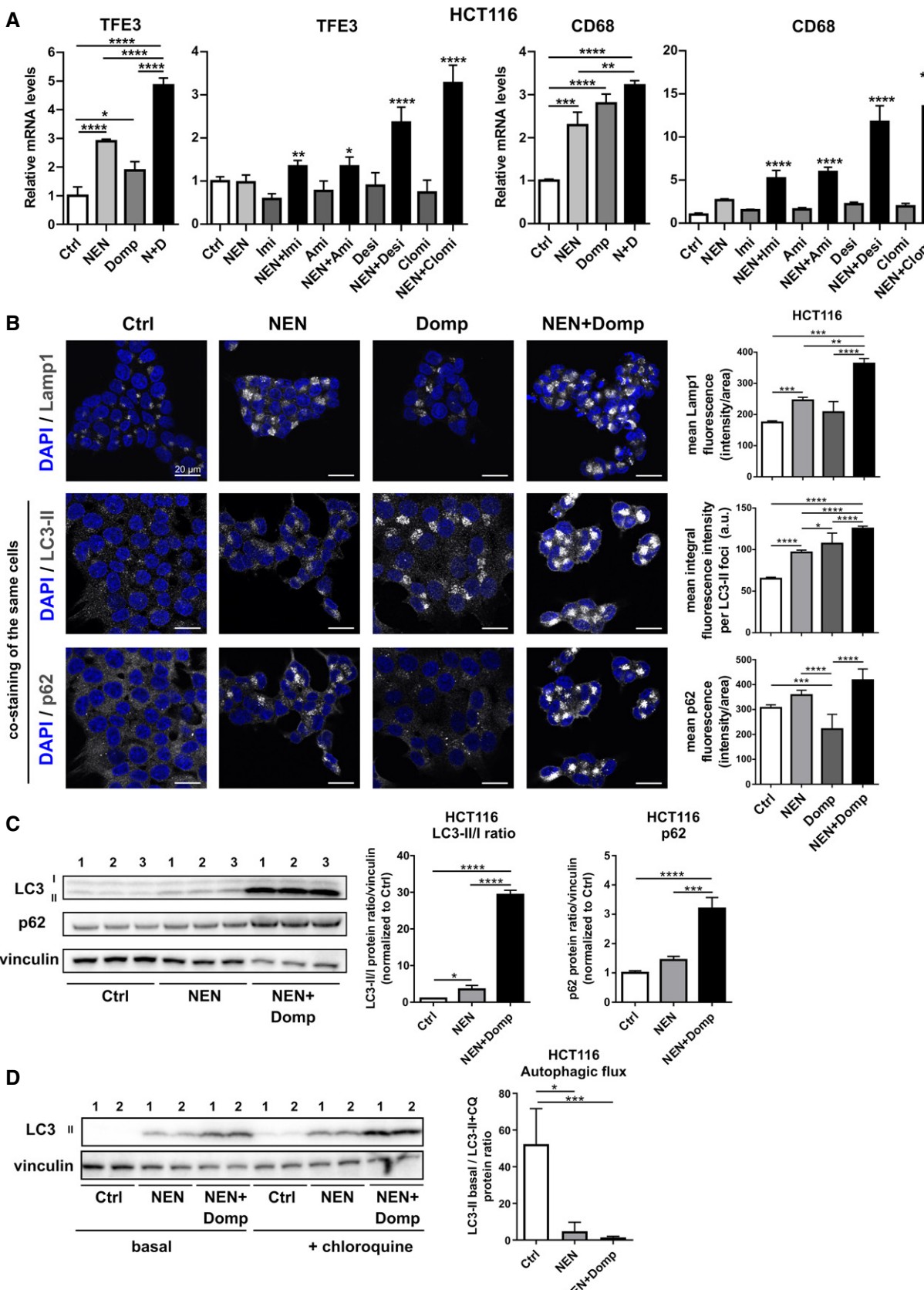

**Figure 3.**

◀

**Figure 3. Combinatorial treatments induces CLEAR network associated with a blockage in autophagy.**

A Relative mRNA expression levels of indicated CLEAR network genes in HCT116 cells treated as indicated ((NEN (N) 1.2 μM; domperidone (Domp, D), imipramine (Imi), desipramine (Desi), and amitriptyline (Ami), each 30 μM; clomipramine (Clomi) 20 μM), determined by qPCR.

B Immunofluorescence (IF) of HCT116 cells treated with DMSO (Control (Ctrl)), NEN, domperidone (Domp) or NEN + Domp as indicated (NEN, 1.2 μM; Domp, 30 μM) for 16 h and stained with anti-Lamp1 (top panel), Alexa-555 LC3 (middle panel) co-stained with Alexa-647 p62 (bottom panel). Nuclei, Blue (dapi). Scale bar 20 μm. Mean (SEM) fluorescence of Lamp1 (Ctrl, NEN, Domp ($N$ = 20 cells); NEN + Domp ($N$ = 24 cells)), LC3-II (Ctrl, NEN + Domp ($N$ = 16 cells)); NEN, Domp ($N$ = 16 cells)) or P62 (Ctrl ($N$ = 38 cells); NEN ($N$ = 27 cells); Domp ($N$ = 28 cells); NEN + Domp ($N$ = 21 cells)) was quantified and analyzed by one-way ANOVA with Tukey *post hoc* test (for LC3-II) or Kruskal–Wallis with Dunn's *post hoc* test (for Lamp1 and p62). *$P$ < 0.05; **$P$ < 0.01; ***$P$ < 0.001; ****$P$ < 0.0001.

C Immunoblot analysis and quantification (mean ± SD) of LC3-II/LC3-I ratios and P62 in HCT116 cells. Cultured cells were treated with DMSO (Control), NEN or NEN and domperidone (Domp) as indicated (NEN, 1.2 μM; Domp, 30 μM) for 16 h. Protein band signal intensities were normalized to vinculin (loading control), and the ratios LC3-II bands LC3-I ($N$ = 3) and p62 levels ($N$ = 3) were analyzed by one-way ANOVA with Tukey *post hoc* test. *$P$ < 0.05; ***$P$ < 0.001; ****$P$ < 0.0001.

D Immunoblot analysis and quantification of LC3 in HCT116 cells (mean ± SEM). Cultured cells were treated with DMSO (Control (Ctrl)), NEN or NEN and domperidone (NEN + Domp) as indicated (NEN, 1.2 μM; Domp, 30 μM) for 16 h, followed by control or 20 μM chloroquine treatment for 3 h. Protein band signal intensities were normalized to vinculin (loading control), and the ratios LC3-II bands in presence or absence of chloroquine treatment ($N$ = 6) were analyzed by Kruskal–Wallis with Dunn's *post hoc* test. *$P$ < 0.05; ***$P$ < 0.001.

Data information: In (A) (first (TFE3) and third (CD68) graph ($N$ = 3); second (TFE3) and fourth (CD68) graph ($N$ = 5, besides NEN + Clomi ($N$ = 4) and NEN + Desi ($N$ = 4)), data are presented as mean (SD) and were analyzed by a one-way ANOVA with Tukey *post hoc* test. In the second and the fourth graph, significance is indicated for the comparison of combinatorial treatments (NEN + TCAs) to controls (Ctrl). *$P$ < 0.05; **$P$ < 0.01; ***$P$ < 0.001; ****$P$ < 0.0001. Exact $P$-values for all comparisons are listed in Appendix Table S1.

---

Chloroquine treatments block the acidification of lysosomes and thus induce a block of lysosomal autophagosome clearance. Confirming the initial autophagy data and adding to the observed accumulation of lysosomes upon double treatments, we observed a strong decrease in autophagic flux upon NEN and NEN + Domp treatment in these cells, indicating a reduction in autophagosome clearance (Fig 3D). These data demonstrated that drug-induced activation of the CLEAR network causes an increase in autophagosome formation but at the same time, the lysosomal clearance of autophagosomes is blocked, presumably compromising the protective function of induced autophagy for cell survival.

## Uridine Phosphorylase 1 (UPP1) induction deregulates pyrimidine metabolism and contributes to cell death

To reveal the metabolic contribution to the observed induction of cell death, we performed metabolomics analysis of control HCT116 cells and cells upon single and combinatorial drug treatment. Strikingly, we found a marked accumulation of the pyrimidine nucleosides uridine and cytidine as well as pre-cursors of de-novo pyrimidine biosynthesis, including orotate and dihydroorotate, particularly upon combinatorial drug treatment. In contrast, uridine and cytidine di- and tri-phosphates were reduced in the drug-treated groups, suggesting a decreased energy load (Fig 4A). Interestingly, this metabolite pattern closely resembled the response of yeast cells to carbon starvation (Xu *et al*, 2013; Huang *et al*, 2015). Similar to what we observed in drug-treated cancer cells, yeast cells growing under conditions of carbon restriction show an accumulation of nucleosides, the latter of which are further catalyzed providing ribose which enters the non-oxidative pentose phosphate pathway (PPP). Induction of the PPP results in enhanced NADPH production to cope with oxidative stress (Xu *et al*, 2013). Interestingly, one of the most strongly upregulated mRNAs upon combinatorial drug treatment in the transcriptome analysis shown in volcano blot in Appendix Fig S7 encoded for uridine phosphorylase 1 (UPP1), an enzyme which catalyzes the reaction of uridine to uracil and ribose, representing a central step in uridine catabolism. Notably, the induction of UPP1 upon combinatorial drug treatment was in agreement with the carbon starvation response in yeast cells, which was

also characterized by the upregulation of the corresponding uridine hydrolyzing enzyme Urh1 (Xu *et al*, 2013).

Uridine is a pyrimidine nucleoside that contributes to nucleotides production required for the nucleic acid synthesis. Uridine catabolism results in metabolites that can be further converted to generate energy or shuttled back to pyrimidine salvage. Therefore, UPP1 upregulation might have reflected an overall catabolic cellular response on the level of nucleic acids. Although the degradation of pyrimidines on the one side can be used to gain ribose and uracil for temporary energy production, it can on the other side cause nucleotide imbalance or shortage, potentially contributing to lethal DNA damage. Therefore, we decided to characterize the mechanistic role of elevated UPP1 levels in drug-induced cell death.

Importantly, using qPCR as well as immunoblot analyses we confirmed that combined treatment with NEN and domperidone or TCAs induced a synergistic and significant upregulation of UPP1 expression (Figs 4B and EV4B and C, Appendix Fig S8), which was observed for all hit compounds. Next, we searched for upstream regulators of UPP1. As demonstrated above, combinatorial drug treatments induced the integrated stress response (ISR, Fig 2) as well as the CLEAR network (Fig 3), the latter including increased levels of the transcription factors TFE3 and MITF. Given the central role of these two stress response regulatory programs, we asked if these pathways could also account for the observed induction of UPP1 upon combinatorial drug treatment. Indeed, inhibition of the ISR by ISRIB treatment as well as double knockdown of TFE3/MITF transcription factors both prevented the drug-induced UPP1 upregulation in HCT116 cells, suggesting that both pathways merged in the regulation of UPP1 (Fig 4C). We decided to test whether the upregulation of UPP1 was a compensatory survival mechanism or on the contrary a mediator of cell death induction. To this end, we depleted UPP1 using siRNAs upon validation of their knockdown efficiency under control and combinatorial drug treatment-induced conditions (Fig 4D). We cultivated knockdown or control cells either in 2D cultures or as spheroids and treated the different groups with NEN + domperidone in order to measure cell death induction. Strikingly, UPP1 knockdown cells were widely protected from drug-induced cell death, both under 2D and spheroid culture conditions (Fig 4E), demonstrating that induction of UPP1 was required for

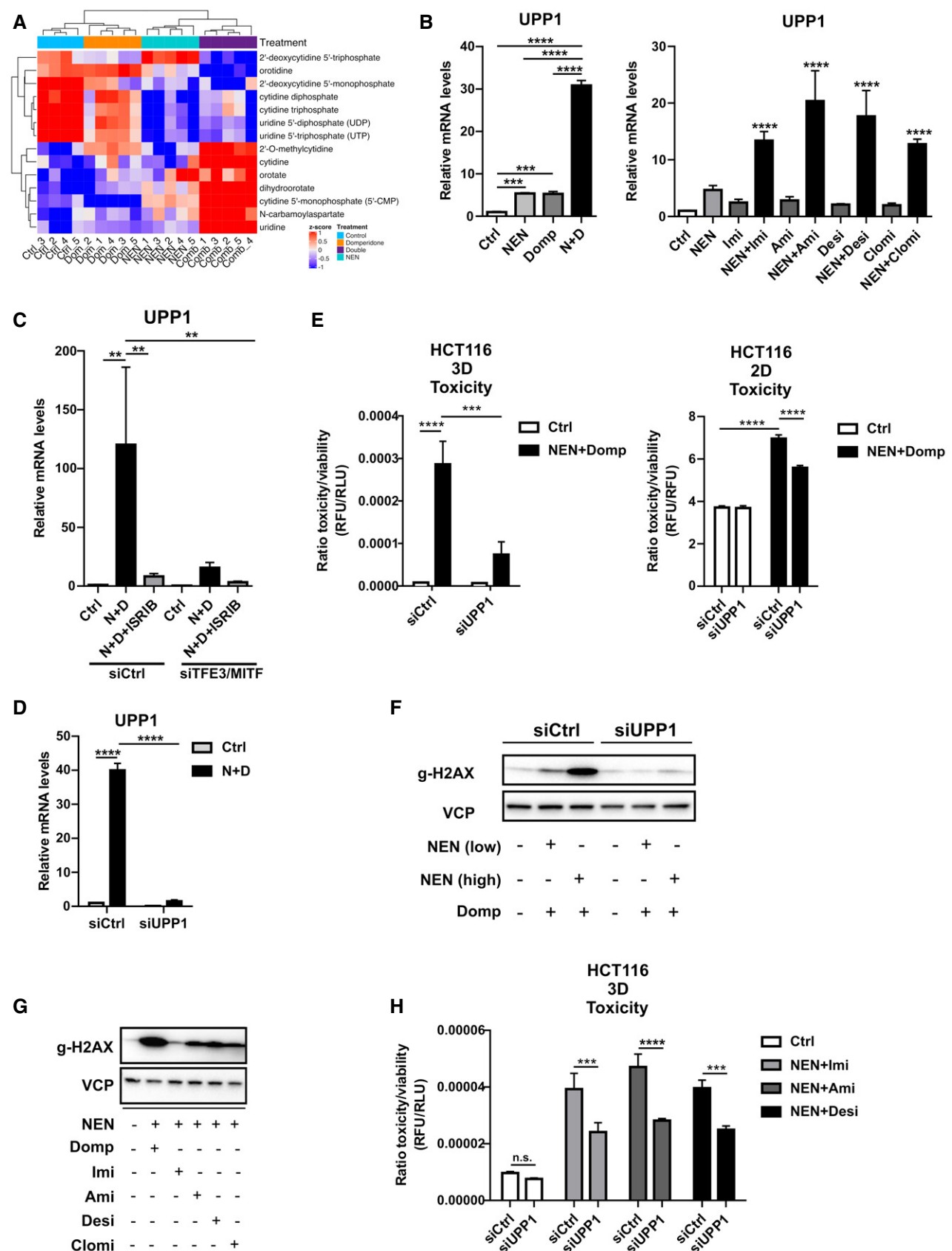

**Figure 4.**

**Figure 4.   UPP1 induction deregulates pyrimidine metabolism and contributes to cell death.**

A   Heat map for pyrimidine pathway-related metabolites derived from a non-targeted metabolomics analysis, depicting relative levels in HCT116 cells treated as indicated for 18 h (NEN (N) 1.2 μM, domperidone (Dom) 30 μM, or in combination (Comb).

B   Relative UPP1 mRNA expression levels in HCT116 cells treated as indicated (NEN (N) 1.2 μM, domperidone (Domp, D) 30 μM, imipramine (Imi), desipramine (Desi), amitriptyline (Ami) each 30 μM, clomipramine (Clomi) 20 μM), determined by qPCR.

C   Relative UPP1 mRNA expression levels in HCT116 cells either transfected with control (siCtrl) or combined TFE3- and MITF-targeting (siTFE3/siMITF) siRNAs and treated as indicated (NEN (N) 1.2 μM, domperidone (D) 30 μM, ISRIB 1 μM), determined by qPCR.

D   Relative UPP1 mRNA expression in HCT116 cells transfected either with control (siCtrl) or UPP1-targeting (siUPP1) siRNAs and treated as indicated (NEN (N) 1.2 μM, domperidone (D) 30 μM).

E   Ratio of toxicity and viability measurements of HCT116 spheroids (3D) and 2D cultures, treated as indicated (NEN 1.2 μM, domperidone 30 μM). Cells were transfected either with control (siCtrl) or UPP1-targeting (siUPP1) siRNAs prior to spheroid formation or during cell seeding.

F   Immunoblot of HCT116 whole cell lysate using antibodies against the indicated proteins upon treatment of the cells as indicated for 16 h (NEN (low) 0.6 μM or NEN (high) 1.2 μM, domperidone (Domp) 30 μM. Cells were transfected either with control (siCtrl) or UPP1-targeting (siUPP1) siRNAs 48 h prior to treatment.

G   Immunoblot of HCT116 whole cell lysates using antibodies against the indicated proteins upon treatment of the cells as indicated for 16 h (NEN 1.2 μM, domperidone (Domp), imipramine (Imi), amitriptyline (Ami) and desipramine (Desi), each 30 μM, clomipramine (Clomi) 20 μM).

H   Ratio of toxicity and viability of HCT116 spheroids (3D) treated as indicated (NEN 1.2 μM, imipramine (Imi), desipramine (Desi), amitriptyline (Ami), each 30 μM). Cells were transfected with control (siCtrl) or UPP1-targeting (siUPP1) siRNAs 24 h prior to spheroid formation.

Data information: In (B, C, D, E and H), data (N = 3) are presented as mean (SD) and were analyzed by one-way ANOVA with Tukey *post hoc* test (B) or two-way ANOVA with Tukey *post hoc* test (C, D, E, H). In (B), right panel, significance is indicated for the comparison of combinatorial treatments (NEN + TCAs) to controls. **$P < 0.01$; ***$P < 0.001$; ****$P < 0.0001$. Exact $P$-values for all comparisons are listed in Appendix Table S1.

---

drug-induced cell death. We observed an increase in the DNA damage marker gamma-H2AX in control cells upon combinatorial drug treatment, which suggested an induction of DNA damage. In contrast, UPP1 knockdown prevented DNA damage induction in agreement with the observed enhanced survival (Fig 4F). In line with induced DNA damage resulting from enhanced pyrimidine degradation, we observed an increase in the DNA damage marker gamma-H2AX upon treatment of HCT116 cells with the different drug combinations (Fig 4G), as well as reduced toxicity upon knockdown of UPP1 (Fig 4H). We confirmed these results in BxPC3 cells, which also showed synergistically induced UPP1 expression levels and DNA damage upon combinatorial drug treatments (Fig EV4A and D). Furthermore, UPP1 knockdown prevented the drug-induced DNA damage and partially suppressed drug toxicity (Fig EV4E and F). Together, these results demonstrated that the induction of UPP1 was crucial for drug efficacy to induce cell death and suggested that nucleotide degradation did not represent a compensatory mechanism to support cell viability under these treatment conditions. Our findings supported a model in which the cellular response comprised an integrated activation of the ISR and CLEAR network, representing a pathway crosstalk that elicits a starvation-like response including enhanced degradation of nucleotides and upregulation of UPP1 contributing to cell death.

### Drug-induced downregulation of pyrimidine biosynthesis enzyme DHODH contributes to cell death

Inversely to the increase in UPP1 expression, we observed a decreased expression in the pyrimidine biosynthesis enzyme dihydroorotate dehydrogenase (DHODH) upon single and combinatorial treatment (Fig 5A). This suggested an attenuated biosynthesis of pyrimidines in addition to the increased degradation of the present pool of nucleosides. To test if the downregulation of DHODH contributed to cell death upon combinatorial treatment, we exposed the cells to sub-lethal drug concentrations and treated additionally with the DHODH inhibitor A771726 (Williamson *et al*, 1995). Strikingly, low-dose NEN + domperidone or TCAs sensitized to A771726 treatment, as indicated by significantly increased cell death in triple-

treated cells, while double and single treatment did not induce cell toxicity (Fig 5B and D). Interestingly, concomitant JNK inhibition, as a mediator of cell death, prevented apoptosis under triple treatment conditions (Fig 5B). Accordingly, the sub-lethal NEN + domperidone treatment led to a slight increase in the phosphorylation of H2AX, whereas the triple combination led to a strong increase in DNA damage (Fig 5C). Taken together, these results suggested that our combinatorial treatment induced cell death by interference with pyrimidine homeostasis, attenuating biosynthesis as well as inducing the degradation via UPP1, overall leading to decreased pyrimidine abundance and cell death.

### Cholesterol dysregulation contributes to stress pathway induction and cell death

We established the CLEAR network and the ISR as two pathways responding to combinatorial drug treatments. However, the upstream signals leading to the induction of these stress response pathways remained unknown. In this regard, we became interested in cholesterol metabolism, as one of the predicted metabolic pathways according to the transcriptome analysis (Appendix Fig S5), since cholesterol dysregulation is able to serve as an inducer of the ISR (Harding *et al*, 2005). In addition, cellular cholesterol transport involves endosomes and lysosomes. Interestingly, due to their chemical nature, TCAs can inhibit lysosomal enzymes, which can lead to transport dysregulation and ultimately cholesterol accumulation (Rodriguez-Lafrasse *et al*, 1990; Wichit *et al*, 2017). Accordingly, cholesterol and Lamp-1 co-immunostaining revealed drug-induced accumulation of cholesterol in late endosomal/lysosomal compartments (Fig 6A and B), which was in agreement with the observed blockage of autophagy at the level of lysosomal degradation (Fig 3). To test whether this accumulation was of functional importance, we made use of Cyclodextrin (CD), a pre-clinically tested substance for Niemann–Pick disease mouse models. NPC is a disorder caused by mutation in the NPC1 gene, which leads to aberrant accumulation of cholesterol in the lysosomes of patients. Cyclodextrin acts a solubilizer and removes cholesterol from the lysosomes (Yu *et al*, 2014; Singhal *et al*, 2018). Co-treatment of

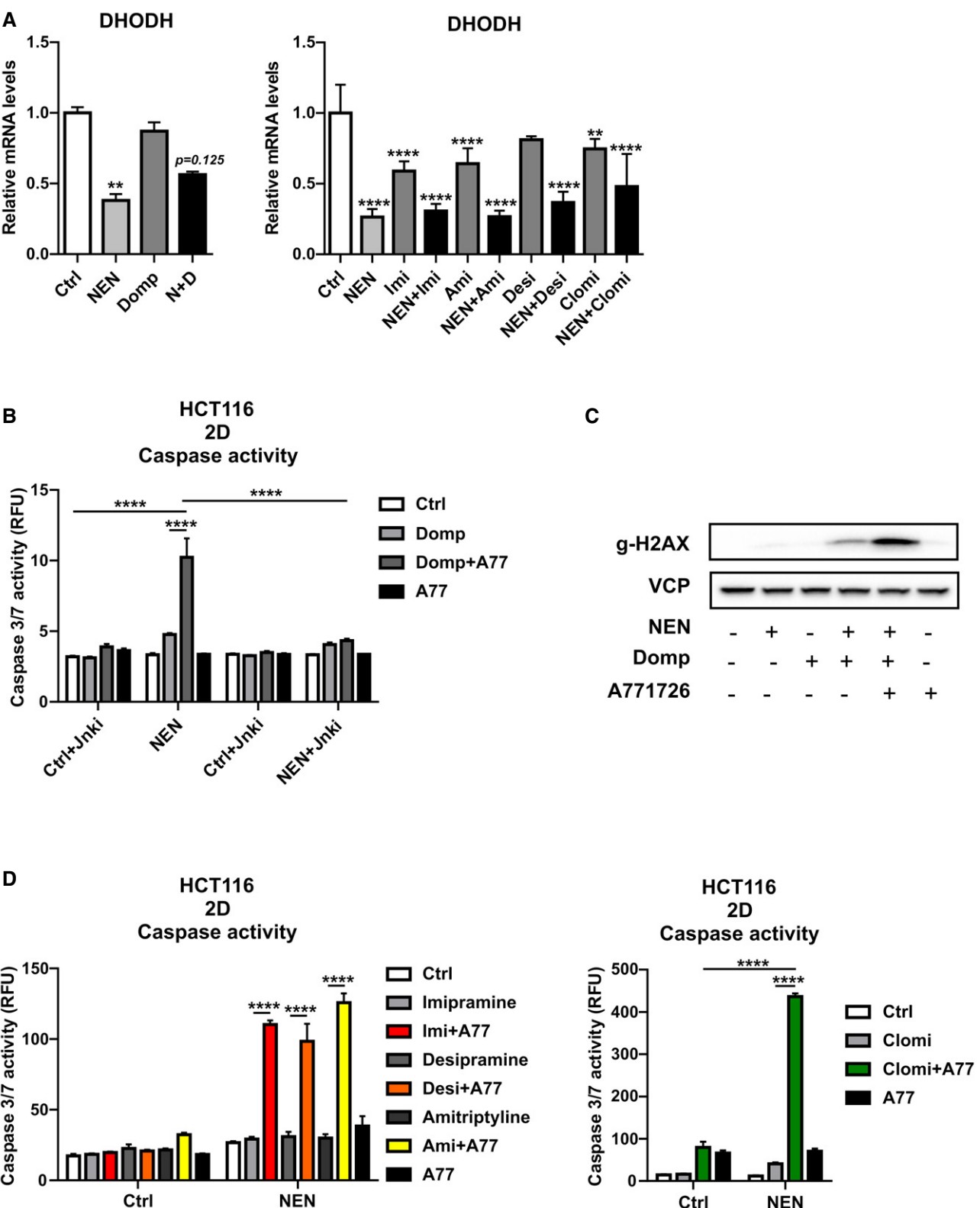

**Figure 5.**

HCT116 spheroids with NEN + domperidone or TCAs and CD resulted in a CD concentration-dependent suppression of cell death (Fig 6C). CD also partially prevented cell death induction upon treatment with our drug combinations in U87 spheroids (Fig 6D). Moreover, we observed partial reversion in DNA damage marker induction and ISR and CLEAR gene expression upon cholesterol

**Figure 5. Drug-induced regulation of pyrimidine biosynthesis enzyme DHODH contributes to cell death.**

A   Relative mRNA expression levels of DHODH in HCT116 cells treated as indicated for 16 h (NEN (N) 1.2 μM, domperidone (Domp, D), imipramine (Imi), desipramine (Desi), amitriptyline (Ami), each 30 μM, clomipramine (Clomi) 20 μM).
B   Caspase activity in HCT116 cells treated as indicated for 24 h (NEN 0.6 μM, domperidone (Domp) 20 μM, A77126 (A77) 50 μM, JNK inhibitor II (Jnki) 1 μM).
C   Immunoblot analysis of HCT116 whole cell lysates using antibodies against indicated proteins and treated as follows: NEN and domperidone for 12 h, A77126 for an additional 6 h (NEN 0.6 μM, domperidone (Domp) 20 μM, A771726 50 μM).
D   Caspase activity in HCT116 cells treated as indicated for 24 h (NEN 0.6 μM; imipramine (Imi), desipramine (Desi), and amitriptyline (Ami), each 20 μM; clomipramine (Clomi) 10 μM; A77126 (A77) 50 μM).

Data information: In (A, B and D), data are presented as mean (SD) ($N = 3$) and were analyzed by Kruskal–Wallis with Dunn's *post hoc* test (A, left graph), one-way ANOVA with Tukey *post hoc* test (A, right graph), or two-way ANOVA with Tukey *post hoc* test (B, D). In (A), significance is indicated for the comparison of the different treatments to controls. **$P < 0.01$; ****$P < 0.0001$. Exact *P*-values for all comparisons are listed in Appendix Table S1.

removal (Fig 6E and F). These data showed that drug-induced cholesterol dysregulation contributed to the induction of stress responses and cell death.

### Drug combinations induce toxicity in PDAC patient-derived organoids (PDOs) and sensitize to paclitaxel treatment

To investigate the efficacy of targeting the described metabolic stress response in a more heterogeneous cancer cell population better reflecting tumor biology (Boj *et al*, 2015; Moreira *et al*, 2018; Tiriac *et al*, 2018; Driehuis *et al*, 2019), we used two patient-derived organoid (PDO) cultures from pancreatic ductal adenocarcinoma (PDAC) patients. PDOs were generated for the biobank at Klinikum Rechts der Isar, Technical University of Munich. PDO-42 was generated from endoscopic ultrasound-guided fine needle aspirations (EUS-FNAs), and PDO-48 was generated from surgical resection. The isolation process of PDOs was mostly resembling what has been described previously (Ruess *et al*, 2018; Renz *et al*, 2018a; Renz *et al*, 2018b; Biederstädt *et al*, 2020; Dantes *et al*, 2020). Similar to spheroid cultures from cell lines, we observed an additive to synergistic effect on cellular toxicity in single and combinatorially treated organoids (Fig 7A). Strikingly, when combined with the chemotherapeutic drug paclitaxel, we observed significant sensitization to chemotherapy in combinatorially treated organoids (Figs 7B and EV5A). In triple combination with NEN and TCAs, paclitaxel was approximately ten times more effective in inducing organoid cell death in PDO-48 and even up to 100 times more in PDO-42 organoids. Also, mRNA expression levels of CHOP and UPP1 were induced upon combined NEN and amitriptyline, resembling the response observed in cancer cell lines (Fig EV5B). Given the excellent safety profile of NEN and TCAs and lack of effective long-term therapeutic options for pancreatic

cancer patients, repurposing of clinical approved drugs for chemotherapeutic sensitization as demonstrated by our study represents an attractive therapeutic strategy.

## Discussion

Due to known toxicity profiles and the undergone drug development process of established drugs, repurposing is a promising strategy to identify effective drugs and drug combinations for novel entities such as cancer. However, often the precise mechanism of how repurposed drugs target cancer cells is not well understood. This is also true for the mitochondrial uncoupler NEN and several TCAs identified in this study to exert combined activity against cancer cells. Different previous studies showed that NEN and niclosamide exert inhibitory effects on tumor growth, which was proposed to be mediated through inhibition of Stat3 transcription factor activity as well as probably indirect negative effects on mTORC1 and Wnt signaling (Arend *et al*, 2016; Alasadi *et al*, 2018). In addition, TCAs were recently shown to be able to induce autophagy via unknown mechanisms (Jeon *et al*, 2011; Gulbins *et al*, 2018). In this respect, our findings importantly contribute to the mechanistic understanding of the individual and combined anti-cancer effects of TCAs and NEN. We characterized a previously unknown function of TCAs and NEN in the synergistic induction of general autophagy as well as specific pyrimidine-degradative pathways mediating toxicity against cancer cells. Additionally, we identified the dopamine receptor antagonist domperidone to synergize with mitochondrial uncouplers in inducing cancer cell death. Furthermore, we provide insights into the upstream regulatory factors and identify CHOP and TFE3 as responsible drivers of the observed degradation pathways.

**Figure 6. Cholesterol dysregulation contributes to stress pathway induction and cell death.**

A   Immunofluorescence (IF) images of HCT116 cells treated as indicated for 12 h (imipramine and domperidone, each 30 μM) and stained for cholesterol content using filipin III.
B   Immunofluorescence (IF) images of HCT116 cells treated with domperidone and co-stained for cholesterol and Lamp1.
C   Ratio of toxicity and viability of HCT116 spheroids (3D) treated as indicated (NEN 1.2 μM, domperidone (Domp) 30 μM). Cyclodextrin content in the medium is indicated in percentage (%CD).
D   Ratio of toxicity and viability of U87 spheroids (3D) treated as indicated (NEN 1.2 μM, domperidone (Domp), imipramine (Imi), amitriptyline (Ami), desipramine (Desi), each 30 μM, clomipramine (Clomi 20 μM, Cyclodextrin (CD) 0.75%).
E   Relative mRNA expression levels of the specified genes in HCT116 cells, treated as indicated for 16 h (NEN (N) 1.2 μM, domperidone (D) 30 μM, Cyclodextrin (CyD) 0.75%).
F   Immunoblot analysis of HCT116 whole cell lysates using antibodies against gammaH2AX and treated as indicated for 16 h (NEN 1.2 μM, domperidone (Domp) 30 μM, Cyclodextrin (CD) 0.75%).

Data information: In (C, D and E), data are presented as mean (SD) ($N = 3$) and were analyzed by a one-way ANOVA with Tukey *post hoc* test or a two-way ANOVA with Tukey *post hoc* test (C and D). *$P < 0.05$; **$P < 0.01$; ***$P < 0.001$; ****$P < 0.0001$. Exact *P*-values for all comparisons are listed in Appendix Table S1.

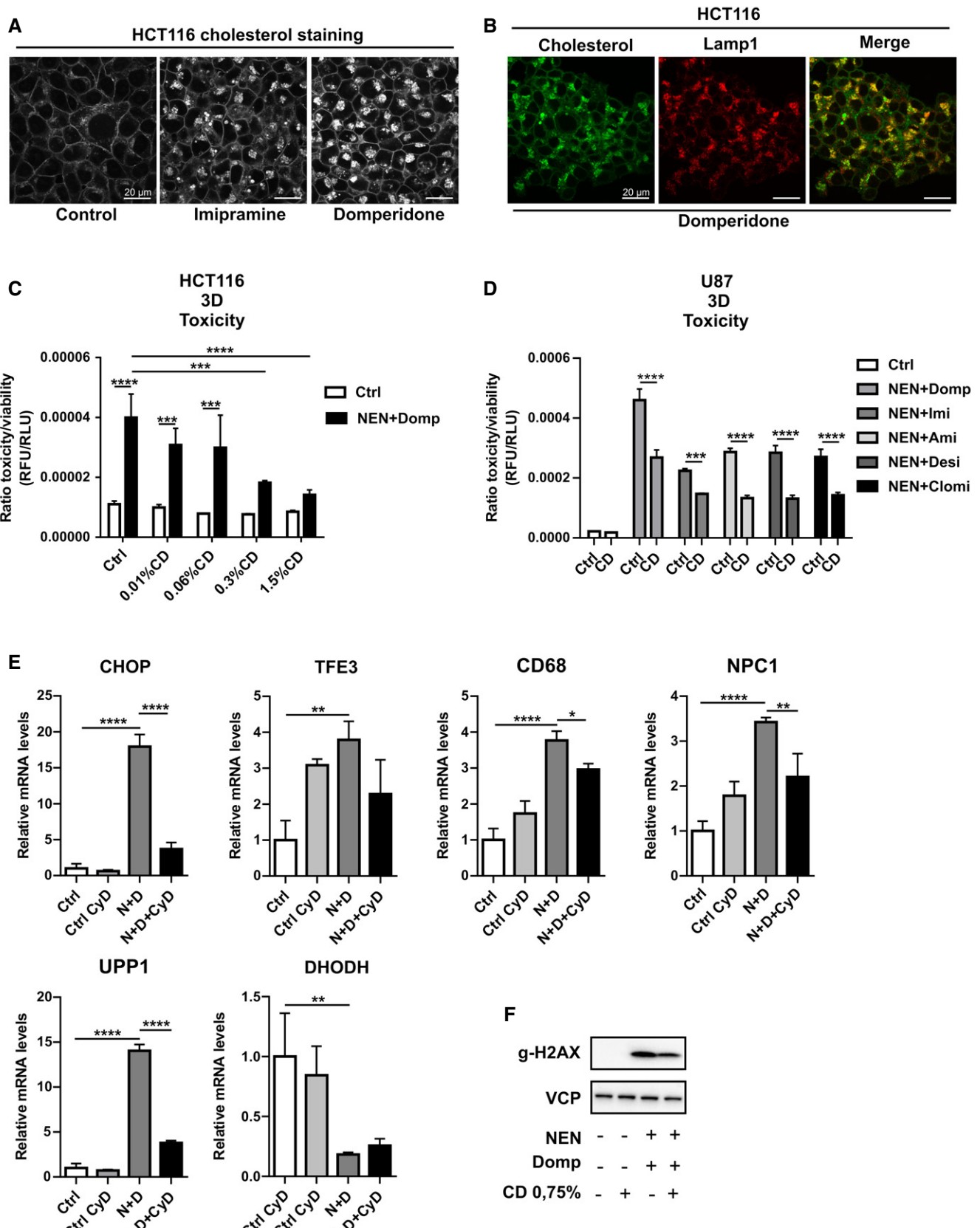

**Figure 6.**

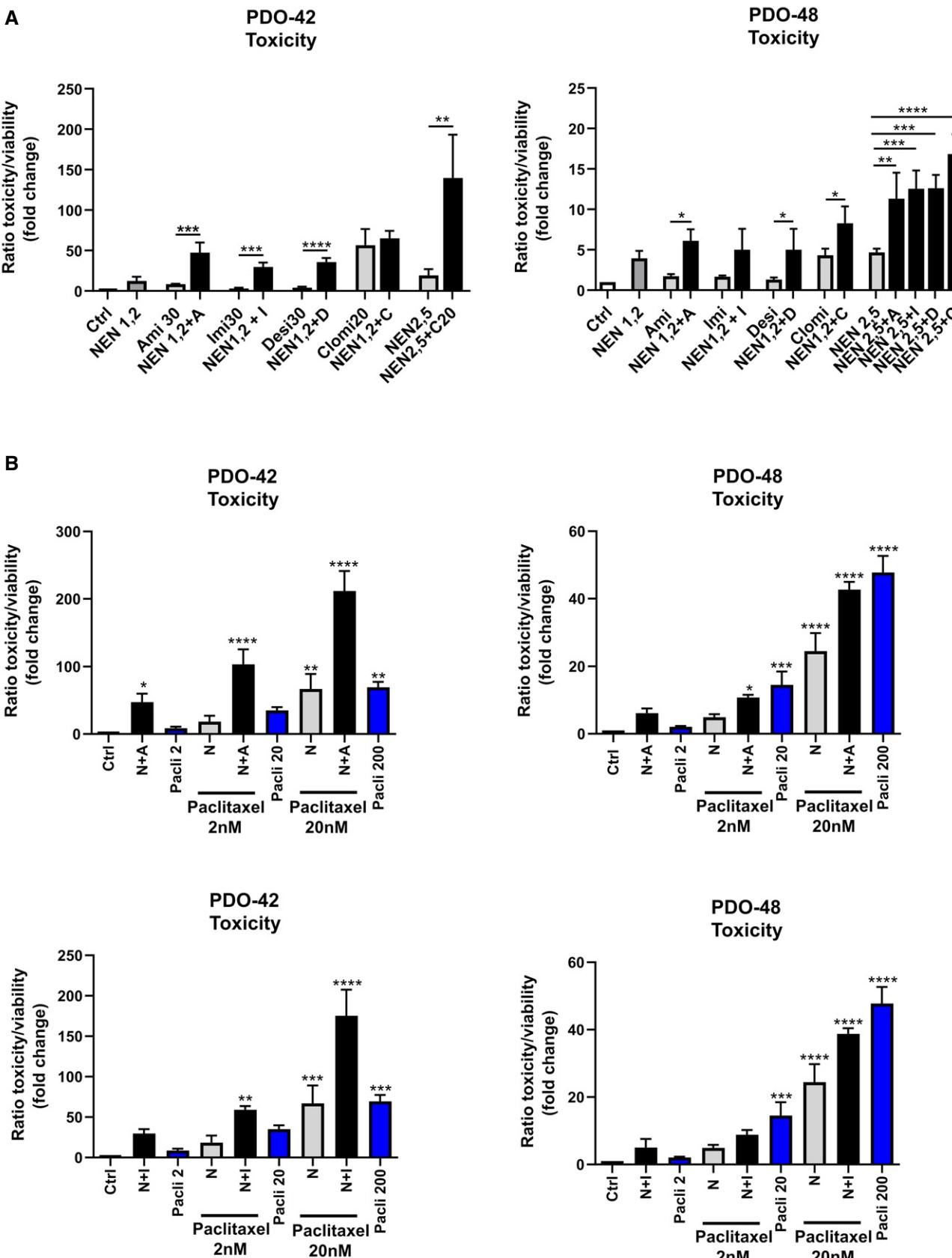

**Figure 7.**

◀

**Figure 7.  Drug combinations induce toxicity in patient-derived organoids and sensitize to paclitaxel treatment.**

A   Ratio of toxicity and viability determined in pancreatic cancer-derived organoids from two patients ((left) PDO-42 and (right) PDO-48) treated as indicated for 3 (PDO-42) or 5 (PDO-48) days. Significant differences are shown for the comparison between single and combined drug treatments as indicated (NEN 1.2 μM or 2.5 μM, amitriptyline (Ami, A), imipramine (Imi, I), desipramine (Desi, D), each 30 μM, clomipramine (Clomi, C) 20 μM).

B   Ratio of toxicity and viability determined in pancreatic cancer-derived organoids from two patients ((left) PDO-42 and (right) PDO-48) treated as indicated for 3 (PDO-42) or 5 (PDO-48) days. Significant differences are shown for the comparison between drug treatments as indicated to the control condition (NEN (N) 1.2 or 2.5 μM, amitriptyline (A), imipramine (I) 20 μM, paclitaxel (Pacli), 2, 20 or 200 nM). As references for sensitization effects to standard chemotherapy, the bars showing single paclitaxel treatments using the indicated concentrations are highlighted in blue. Bars for controls (Ctrl), paclitaxel (Pacli), NEN (N) alone, and combined NEN + paclitaxel treatment are identical in the individual graphs (including respective graphs in Fig EV5A) for comparison to combinations with the respective TCA drugs as indicated.

Data information: In (A and B), data are presented as mean (SD) ($N = 3$) and were analyzed by a one-way ANOVA with Tukey *post hoc* test. *$P < 0.05$; **$P < 0.01$; ***$P < 0.001$; ****$P < 0.0001$. Exact *P*-values for all comparisons are listed in Appendix Table S1.

Since NEN (more specifically niclosamide) and certain TCAs (imipramine and desipramine) are currently or have been in clinical testing as monotherapy for metastatic prostate cancer (niclosamide: NCT02532114), small-cell lung cancer and other neuroendocrine tumors (NCT01719861), and triple-negative breast cancer (NCT03122444), the presented results are of importance in several aspects. First, we here show a synergistic augmentation of anti-cancer efficacy for defined drug combinations highlighting the potential of future combinatorial therapies. This is particularly relevant given that the efficacy of the singularly tested drugs in the clinical setting is not yet confirmed. The TCA imipramine was shown to possess promising efficacy in advanced mouse models of glioblastoma and against small-cell lung cancer (Jahchan *et al*, 2013; Shchors *et al*, 2015) and as mentioned above, a trial for TNBC is currently running. However, the clinical trial with desipramine, the derivative of imipramine, as monotherapy for small-cell lung cancer was terminated due to lack of efficacy (NCT01719861). Although the underlying reasons were not elucidated, a combination therapy like the one we propose might represent a promising strategy to augment efficacy and repurpose other TCAs for cancer therapy in a broader entity context. Importantly, we also showed efficacy in glioma cell lines and pancreatic cell lines as well as primary organoid models, underlining a potential broad efficacy spectrum. Generally, with the exception of domperidone, all drugs have favorable safety profiles and are well tolerated over prolonged periods. Nevertheless, despite hERG channel inhibitory effects of domperidone, it is regularly given as supportive therapy to overcome chemotherapy-induced nausea. In this light, it would be interesting to follow-up patients given domperidone in combination with chemotherapy and test for overall survival rate and progression-free survival in comparison to non-domperidone receivers.

Second, we present a detailed molecular mechanism of the anti-cancer effect of TCAs and NEN, which could represent the basis for the translation into more refined cancer treatment strategies as compared to the ongoing clinical testing. We propose a model in which the combination of NEN with domperidone or TCAs synergistically triggers two central stress response pathways (the ISR and the CLEAR network), which merge in the induction of the pyrimidine-degradative enzyme UPP1 required for cell death upon combined drug treatments. We also observed a downregulation of the pyrimidine biosynthesis enzyme DHODH, suggesting reduced pyrimidine biosynthesis. This lowered expression might be a reason for the observed sensitization to the specific DHODH inhibitor A771726. Interestingly, DHODH inhibitors are currently in clinical trials (NCT03404726), offering the possibility to test synergistic triple treatments with the drug combinations identified here in refined treatment approaches.

Overall, blockage of lysosomal degradation albeit enhanced autophagosome formation, UPP1-dependent induced catabolism, and reduced synthesis of pyrimidines might jointly contribute to cell death. Of note, the induction of autophagosome formation and UPP1 matches a starvation response of yeast cells (Xu *et al*, 2013; Huang *et al*, 2015), indicating a pharmacological starvation mimic function of the identified drug combinations. It is known that cancer cells, which reside distant from blood vessels, undergo extensive metabolic stress and nutrient limitation, which makes them more susceptible to chemotherapy (Schug *et al*, 2015; Kumar *et al*, 2019). This might explain why the tested drug combinations sensitize human pancreatic tumor organoids to the standard-of-care therapy paclitaxel. Sensitization to standard-of-care chemotherapy such as paclitaxel offers the opportunity to facilitate clinical testing of the proposed treatments in order to prevent and/or overcome drug-resistance. Besides the envisioned drug-repurposing approaches, the novel mechanistic insights into metabolic stress-induced cancer cell death enable the identification of novel compounds for the development cancer metabolism-targeting treatment concepts.

# Materials and Methods

### Small molecule screening

Plate and liquid handling was performed using a HTS platform system composed of a Sciclone G3 Liquid Handler from PerkinElmer (Waltham, MA, USA) with a Mitsubishi robotic arm (Mitsubishi Electric, RV-3S11), a MultiFlo™ Dispenser (BioTek Instruments, Bad Friedrichshall, Germany), and a Cytomat™ Incubator (Thermo Fisher Scientific, Waltham, MA, USA). Cell seeding and assays were performed in black 384-well plates (Greiner bio one 384-well μCLEAR®, BLACK, 781091). The plates were coated with poly-D-lysine (Sigma-Aldrich, St. Louis, MA, USA) for 1 h at room temperature to facilitate a better cell adherence. Cells were seeded in 384-well microplates with a cell number of 12,000 cells/ well. The signals of the MultiTox-Fluor assays were detected on the EnVision® Multilabel Reader (PerkinElmer, Waltham, MA, USA). The Prestwick Chemical library contains 1,280 small molecule compounds, which are 100% approved drugs (Food and Drug Administration (FDA), European Medicines Agency (EMA) and other agencies). The purity of the compounds was > 90% as reported by the provider of the compounds. The GPCR compound library was purchased from MedChem Express (Cat. No.: HY-L006). The Z' factor was calculated as described by Zhang and colleagues according to the formula

$Z' = 1 - (3(\theta p + \theta n)/(\mu p - \mu n))$, where $p$ is the positive control, $n$ is the negative control, $\theta$ is the standard deviation, and $\mu$ is the mean. (Zhang *et al*, 1999). For hit selection, a threshold of lower than 3 standard deviations from the median of the negative/ vehicle population was set.

## Screening assay

Before conducting drug screening, we determined the optimal concentrations for NEN and the positive control in several assay development experiments. We have hereby defined 1.2 μM NEN as the optimal concentration, since the cells only show minor increased toxicity at this concentration, but are sensitized to the positive control. The small molecules of the Prestwick and GPCR library were screened at ~10 μM in order to keep the concentration of the solvent DMSO at a maximum of 1%. A concentration of 10 μM is an accepted standard approach for primary screening campaigns (Hughes *et al*, 2011). The substances of the Prestwick and GPCR library usually show no or only mild toxic effects within a period of 24 h at a concentration of 10 μM and were therefore chosen as appropriate condition for the detection of synergistic toxicity effects in combination with NEN.

For screening of the FDA and GPCR small molecule libraries, HCT116 cells were washed with PBS, trypsinized, and resuspended in cell culture medium. The cell suspension (12,000 cells in 50 μl per well) was dispensed into poly-D-lysine pre-coated 384-well plates (Greiner bio one 384-well μCLEAR®, BLACK, 781091) and incubated at 37°C with 5% $CO_2$. 24 h after seeding, either 5 μl of NEN (negative control), NEN + Nutlin-3a (positive control), or DMSO alone (solvent control) dissolved in cell culture media were dispensed to the assay plate (NEN, final concentration: 1.2 μM; Nutlin-3a, final concentration: 25 μM). In addition, the same day cells were treated either with compound (1 mM stock solution) dissolved in 100% DMSO or DMSO alone. 0.5 μl of compounds/ DMSO were transferred to 55 μl cell culture medium per well to keep the final DMSO volume concentration below 1.0%. The cells were then incubated (37°C; 5% $CO_2$) for 24 h prior to adding 50 μl MultiTox-Fluor Multiplex Cytotoxicity Assay reagent (Promega). Live-cell fluorescence (excitation = 400 nm; emission = 505 nm) and dead-cell fluorescence (excitation = 485 nm; emission = 520 nm) were recorded with an EnVision Multilabel Reader (PerkinElmer, Waltham, USA). The ratio for live/dead was calculated and normalized to the negative/vehicle control. For hit selection, a threshold of lower than 3 standard deviations from the median of the negative/ vehicle population was set.

## Generation and dissociation of pancreatic cancer patient-derived organoids (PDOs)

PDAC patient-derived organoids were either generated from surgical resection specimen (PDO-48) or endoscopic ultrasound-guided fine needle aspiration (PDO-42). PDAC was confirmed by histopathology. Based on the mutational profile, PDO-48 and PDO-42 are tumor-derived. The step-by-step procedure of PDO generation and single cell dissociation for further drug testing has been described (Dantes *et al*, 2020). In brief, resection specimens were dissected into small fragments and washed with washing media (Advanced DMEM/F12 with 1× GlutaMax, 10 mM HEPES, and 100 μg/ml Primocin). Next, red

blood cells (RBCs) were eliminated by incubation of the sample with RBC lysis buffer (ACK lysis buffer, Life Technologies) for 10–15 min at room temperature. After washing and centrifugation (5 min, 4°C, and 200 $g$), the resection sample was subjected to enzymatic digestion by collagenase (collagenase type II, Life Technologies) at 5 mg/ml, whereas the FNA samples were digested by a short incubation (5–10 min) with TrypLE (Life Technologies) at 37°C. Next, the primary cells were mixed with Matrigel and plated as domes of Matrigel in a 24-well plate. After solidification of the Matrigel dome, cells were supplemented with warm organoid feeding media (Boj *et al*, 2015; Dantes *et al*, 2020). For further drug testing, PDOs were dissociated into single cells as recently described (Biederstadt *et al*, 2020). Mechanical and enzymatic digestion (Dispase II and/or TrypLE) of PDOs into single cells was confirmed microscopically. After counting the single cells, 1,000 cells per well (in a mixture of 80 μl organoid feeding media and 10 μl Matrigel) were plated in a white 96-well plate (Corning, Cat. No. 3610). Each well was pre-coated with a 30-μl mixture of Matrigel and PBS (1:3 or 1:4 ratio).

All patients enrolled in the study were consented prior to investigation based on the institutional review board (IRB) project-number 207/15 of the Technical University Munich. All experimental procedures were performed in agreement with the ethical principles for medical research involving human subjects as defined by the WMA Declaration of Helsinki and the Department of Health and Human Services Belmont Report.

## Cell collection and homogenization for non-targeted metabolomics

The medium was aspirated, the cells were quickly washed twice with 2 ml warm PBS, and their metabolism was subsequently quenched by the addition of pre-cooled (dry ice) 400 μl extraction solvent, a 80/20 (v/v) methanol/water mixture which contained four standard compounds for monitoring the efficiency of the metabolite extraction. Cells were scraped off the culture vessel using rubber tipped cell scrapers (Sarstedt) and together with the solvent collected into pre-cooled micro tubes (2.0 ml, Sarstedt). The culture well was rinsed with another 100 μl extraction solvent, and the liquid was also transferred to the tube. Two culture wells were pooled to make one sample. The samples were stored at −80°C until metabolomics analysis.

For homogenization, 160 mg glass beads (0.5 mm, VK-05, Peqlab) were added to the cell samples, which were homogenized using the Precellys24 homogenizer at 0–4°C for two times over 25 s at 5,500 rpm with 5 s pause interval. To normalize the metabolomics data from cell homogenates for differences in cell number, the DNA content was determined using a fluorescence-based assay for DNA quantification. The assay was performed as previously described (Muschet *et al*, 2016). Briefly, the fluorochrome Hoechst 33342 (10 mg/ml, Thermo Fisher Scientific) was diluted into PBS to the final concentration of 20 μg/ml. 80 μl of this dilution was applied to each well of a black 96-well plate (Thermo Fisher Scientific). After brief vortexing of the cell homogenates, 20 μl of the sample was added to the Hoechst dilution to gain 100 μl total volume per well and mixed by pipetting. Each sample was applied to the plate in four replicates. 20 μl extraction solvent was used for blank measurements. The plate was incubated at room temperature in the dark for 30 min, and the fluorescence was read using a GloMax Multi Detection System (Promega) equipped with a UV filter

($\lambda_{Ex}$ 365 nm, $\lambda_{Em}$ 410–460 nm). Subsequently, the samples were centrifuged at 4°C and 11,000 $g$ for 5 min and the supernatant was used for non-targeted metabolomics.

### Non-targeted metabolomics

Aliquots of 105 µl of the supernatant were loaded onto four 96-well microplates. Two (i.e., early and late eluting compounds) aliquots were dedicated for analysis by ultra-high-performance liquid chromatography–tandem mass spectrometry (UPLC-MS/MS) in electrospray positive ionization, one for analysis by UPLC-MS/MS in negative ionization and one for the UPLC-MS/MS in negative ionization for polar compounds. Three types of quality control samples were included into each plate: samples generated from a pool of human plasma, samples generated from a small portion of each experimental sample served as technical replicate throughout the data set, and extracted water samples served as process blanks. Experimental samples and controls were randomized across the metabolomics analysis. The samples were dried on a TurboVap 96 (Zymark).

Prior to UPLC-MS/MS analysis, the dried samples were reconstituted in acidic or basic LC-compatible solvents, each of which contained 8 or more standard compounds at fixed concentrations to ensure injection and chromatographic consistency. The UPLC-MS/MS platform utilized a Waters Acquity UPLC with Waters UPLC BEH C18-2.1 × 100 mm, 1.7 µm columns and a Thermo Scientific Q-Exactive high resolution/accurate mass spectrometer interfaced with a heated electrospray ionization (HESI-II) source and Orbitrap mass analyzer operated at 35,000 mass resolution. Extracts reconstituted in acidic conditions were gradient eluted using water and methanol containing 0.1% formic acid, while the basic extracts, which also used water/methanol, contained 6.5 mM ammonium bicarbonate. The aliquot for polar compounds determination was analyzed via negative ionization following elution from a HILIC column (Waters UPLC BEH Amide 2.1 × 150 mm, 1.7 µm) using a gradient consisting of water and acetonitrile with 10 mM ammonium formate. The MS analysis alternated between MS and data-dependent MS2 scans using dynamic exclusion and a scan range of 80–1,000 $m/z$. Metabolites were identified by automated comparison of the ion features in the experimental samples to a reference library of chemical standard entries that included retention time, molecular weight ($m/z$), preferred adducts, and in-source fragments as well as associated MS spectra and curation by visual inspection for quality control using software developed at Metabolon.

### Transcriptome analysis

For the transcriptome analysis, HCT116 cells were treated with vehicle control, 1.2 µM NEN, 30 µM domperidone, and NEN + domperidone. Cells were harvested after 16 h, before the occurrence of major cell death. Total RNA was isolated employing the RNeasy Mini kit (Qiagen) including on-column DNase digestion. The Agilent 2100 Bioanalyzer was used to assess RNA quality, and only high-quality RNA (RIN > 7) was used for microarray analysis. 3 out of 5 replicates were chosen for the microarray analysis. Total RNA (150 ng) was amplified using the WT PLUS Reagent Kit (Affymetrix, Santa Clara, US). Amplified cDNA was hybridized on Human Clariom S HT arrays (Affymetrix). Staining and scanning was done according to the Affymetrix expression protocol. Transcriptome Analysis Console

(TAC; version 4.0.0.25; Thermo Fisher Scientific) was used for quality control. Array data have been submitted to the GEO database (https://www.ncbi.nlm.nih.gov/geo/) at NCBI (accession: GSE148682).

### Bioinformatic analysis of omics data

Transcriptomics and metabolomics analysis were conducted in R (R Core Team (2019), R: A language and environment for statistical computing. R Foundation for Statistical Computing, Vienna, Austria. http://www.r-project.org/index.html). Affymetrix background correction, quantile normalization, and summarization were done with the package oligo (version 1.50.0) (Carvalho & Irizarry, 2010). Differential gene expression was estimated using the package limma (version 3.42.2) (Ritchie *et al*, 2015). *P*-values were corrected for multiple testing using Bonferroni correction method. Gene set enrichment analysis (GSEA) for GO terms (biological process) and WikiPathways were done using the package clusterProfiler (version 3.14.3) (Yu *et al*, 2012). GSEA results were visually represented with bubble plots created using the package ggplot2 (H. Wickham. ggplot2: Elegant Graphics for Data Analysis. Springer-Verlag New York, 2016). Metabolic data were normalized with the "variance stabilizing normalization" (vsn) method (Huber *et al*, 2002), and missing values were imputed with the k-nearest neighbor (knn) algorithm. Metabolites not present in at least 70% of the samples were removed from further analysis. Differentially expressed metabolites (DEMs) were detected via a one-way ANOVA and a Tukey *post hoc* test (Fig 4A). Heat maps of DEGs and DEMs (Fig 4A and Appendix Fig S1B) were created using the R package ComplexHeatmap (version 2.2.0) (Gu *et al*, 2016).

### Cell culture

HCT116 (human colon carcinoma, (ATCC® CCL-247™)), BxPC3 (human pancreas adenocarcinoma (ATCC® CRL-1687™), and U87 (human glioblastoma (ATCC® HTB-14™)) cell lines were maintained in Dulbecco's modified Eagle's medium (DMEM) (Gibco), supplemented with 10% fetal bovine serum (FBS; Millipore) and penicillin/streptomycin (Life Technologies), and cultured at 37°C, 5% $CO_2$, and 95% humidity. For spheroid formation, 10,000 cells were plated in ultra-low attachment plates (Fisher Scientific), centrifuged at 850 $g$ for 10 min, and incubated for 3 days until treatment, with medium change every 3 days. The absence of contamination with mycoplasma was ensured by regular testing using the PCR Mycoplasma Test Kit I/C (PromoKine PK-CA91-1048). Cell authentication is ensured by the supplier (ATCC).

### Primary hepatocyte isolation

Primary hepatocytes were isolated through collagenase perfusion of 10-week-old male C57BL/6N mice. Mice were anaesthetized and then both abdominal walls were opened. The liver was perfused through the venae cavae with EGTA-containing HEPES/KH buffer for 10 min, followed by a collagenase-containing HEPES/KH buffer for 15 min until liver digestion was visible. The perfused liver was cut out and placed into a suspension buffer-containing dish, and hepatocytes were gently washed out. After filtering the cells through a 100-nm pore mesh, cells were centrifuged and washed twice and resuspended in

suspension buffer. 30,000 cells per well were plated in collagen-coated 96-well plates (Thermo Fisher Scientific) in William E Medium (PAN Biotech) containing 10% FBS (PAN Biotech), 5% penicillin/strepto-mycin (Life Technologies) and 100 μM dexamethasone (Sigma-Aldrich Chemie) and maintained at 37°C and 5% $CO_2$. For the cyto-toxicity test in primary mouse hepatocytes, Apo-One Homogeneous Caspase-3/7 Activity Assay Kit was employed as described in the provider's protocol. Briefly, cultured hepatocytes were treated with the different drug combinations for 24 h before assaying. Subse-quently, the Apo-One Caspase-3/7 reagent mixture was added to cultured cells to a 1:1 volume ratio and incubated on a plate shaker at 300 rpm for 3 h at room temperature in the darkness. The fluores-cence of each well was measured by spectrofluorometer (Thermo Scientific, VARIOSKAN LUX Reader, VLBL0TD2-3020).

### Animals

All animal studies were conducted in accordance with German animal welfare legislation. 10-week-old male C57BL/6N mice used for primary hepatocyte isolation were obtained from the Charles River Laboratories and maintained in a climate-controlled environ-ment with specific pathogen-free conditions under 12-h dark–light cycles in the animal facility of the Helmholtz Center, Munich, Germany. Protocols were approved by the institutional animal welfare officer, and the necessary licenses were obtained from the state ethics committee and government of Upper Bavaria (nos. 55.2-1-55-2532-49-2017 and 55.2-1-54-2532.0-40-15). Mice were fed ad li-bitum with regular rodent chow.

### Chemical compounds

Niclosamide ethanolamine was purchased from Cayman Chemical; domperidone, imipramine, amitriptyline, desipramine, and clomi-pramine were purchased from Sigma-Aldrich.

### siRNA transfection

For siRNA-mediated knockdown experiments, cells were reverse-transfected in 6 wells using Lipofectamine RNAiMAX (Thermo Fisher) mixed with 20nM Dharmacon SMARTpool siRNAs, follow-ing the manufacturer's protocol. After 12-h incubation, cells were trypsinized and seeded for corresponding experiments.

### Immunoblot analyses

Cultured cells were lysed at 4°C in 100 μl of RIPA buffer (Sigma-Aldrich) containing protease (Roche 11697498001) and phosphatase (Roche 04906837001) inhibitors, and sonicated. Cell extracts were centrifuged 18,000 $g$ for 5 min at 4°C and the supernatant collected to protein concentration determination with BCA protein assay (Pierce 23225). 10–20 μg of total protein was subjected to SDS–PAGE, and after transfer, membranes were blocked (30 min) and incubated overnight at 4°C with primary antibodies. After secondary antibody incubation for 1 h at room temperature, protein bands were visualized with Amersham ECL prime and using an ChemiDoc Image System (Bio-Rad) and ImageLab software. For the densito-metric quantification of protein band signal intensities, ImageJ (National Institutes of Health) was used as described previously

(https://lukemiller.org/index.php/2010/11/-analyzing-gels-and-we stern-blots-with-image-j/).

The antibodies used and the respective dilutions are listed in Table 1.

### Crude fractionation of nuclei

Cell pellet was homogenized with 2 ml cold lysis/extraction buffer (20 mM Tris–HCl pH 7.4, 0.25 M sucrose, 1 mM EDTA, 1 mM DTT, +protease inhibitor cocktail) using glass a douncer, followed by centrifugation at 200 G for 30 min at 4°C. Supernatant was collected (total lysates) and pellet discarded. 1 ml aliquot was collected as a control (WL). The remaining lysate was centrifuged at 800 $g$ for 30 min at 4°C. The supernatant was collected as a fraction of all other organelles except nuclei (Cyto), and pellet was resuspended as a crude nuclei fraction (Nu).

### qPCR

RNA was isolated using the Qiagen RNAesay kit following the manufacturer's protocol and reverse-transcribed using the Quanti-Tect Reverse Transcription Kit (Qiagen). cDNA was amplified using the PowerUp SYBR Green Master Mix (Life Technologies), using gene-specific primers (Table 2) and an Applied Biosystems QuantS-tudio 6 cycler. RNA expression data were normalized to the level of beta-actin expression.

### Cell viability measurements

To determine drug-induced toxicity in spheroid cultures, the Promega CellToXGreen and RealTime-Glo™ MT Cell Viability kits were used according to the manufacturer's protocols. Briefly, toxic-ity and viability dyes were mixed with cell culture medium and added prior to drug treatment. Fluorescence (relative florescence units, RFU) and luminescence (relative luminescence units, RFU) were measured over the course of 3 days on a Thermo Fisher Varioskan Lux plate reader.

For Caspase 3/7 activity and general cytotoxicity in 2D cell culture, the Apo-One Homogeneous Caspase-3/7 Assay and MultiTox-Fluor Multiplex Cytotoxicity Assay (both Promega) were employed, respectively, according to the manufacturer's protocol. Cells were seeded in 96 wells 1 day prior to treatment and incubated 24–48 h before assaying.

### Quantification of Apoptosis induction by flow cytometry

HCT116 or U87 cells were seeded in 6-well plates and incubated overnight before treatment. They were treated with the correspond-ing drugs for 24 h (HCT116) or 48 h (U87). The supernatant of each well was collected to 15mL Falcon tubes. The cells were washed with 2 ml PBS, which was also collected into the Falcon tubes. The cells were then trypsinized and added to the same Falcon tube. They were centrifuged at 200 $g$ for 5 min and washed twice: once with FACS buffer (PBS 1% BSA) and once with 1× Annexin binding buffer (BD Biosciences 556454). The cells were resuspended in 100 μl Annexin binding buffer and stained with 10 μl propidium iodide (PI, BD Biosciences 556463) and 5 μl FITC-Annexin V (BD Biosciences 556419) for 15' on ice. The stained cells were then

**Table 1.** List of antibodies.

| Antibody | Dilution |
|---|---|
| p-Histone H2A. XSer139 (20E3) Cell Signaling 9718 | 1:2,000 |
| LC3A/B (D3U4C) Cell Signaling 12741 | 1:1,000 |
| VCP Abcam ab11433 | 1:10,000 |
| Acetyl Histone H3 Millipore 06-599 | 1:1,000 |
| GAPDH Sigma-Aldrich G8795 | 1:10,000 |
| TFE3 Cell Signaling 14779 | 1:1,000 |
| MITF Cell Signaling 12590 | 1:1,000 |
| CHOP Cell Signaling 2895 | 1:500 |
| p-eIF2α Cell Signaling 9721 | 1:500 |
| eIF2α Cell Signaling 9722 | 1:1,000 |
| ATF4 Cell Signaling 11815 | 1:1,000 |
| UPP1 Abcam ab205031 | 1:1,000 |
| Vinculin Abcam ab155120 | 1:1,000 |
| LC3B Cell Signaling 2775 | for immunofluorescence 1:200, for immunoblot 1:1,000 |
| P62 BD Transduction Lab 610832 | 1:1,000 |
| Lamp1 BD Pharmingen 555798 | for immunofluorescence 1:100 |
| Goat anti-Rabbit IgG (H + L) Highly Cross-Adsorbed Secondary Antibody, Alexa Fluor 555 Thermo Fisher Scientific A21429 | 1:1,000 |
| Donkey anti-Mouse IgG (H + L) Highly Cross-Adsorbed Secondary Antibody, Alexa Fluor 555 Thermo Fisher Scientific A31570 | 1:1,000 |
| Goat anti-Mouse IgG (H + L) Cross-Adsorbed Secondary Antibody, Alexa Fluor 647 Thermo Fisher Scientific A21235 | 1:1,000 |
| Alexa Fluor ™ 488 phalloidin Thermo Fisher Scientific A12379 | 1:800 |
| DAPI (4′,6-diamidino-2-phenylindole) Thermo Fisher Scientific D1306 | 1:10,000 |

**Table 2.** qPCR primer sequences.

| | |
|---|---|
| tfe3 f | CCGTGTTCGTGCTGTTGGA |
| tfe3 r | GCTCGTAGAAGCTGTCAGGAT |
| npc1 f | GTCCAGCGCAGGTGTTTTC |
| npc1 r | GCCGAACATCACAACAGAGAC |
| cd68 f | GGAAATGCCACGGTTCATCCA |
| cd68 r | TGGGGTTCAGTACAGAGATGC |
| SQSTM f | GCACCCCAATGTGATCTGC |
| SQSTM r | CGCTACACAAGTCGTAGTCTGG |
| MITF f | GCCTCCAAGCCTCCGATAAG |
| MITF r | CATCTGCTCACGCATGAGTTG |
| dhodh f | CCACGGGAGATGAGCGTTTC |
| dhodh r | CAGGGAGGTGAAGCGAACA |
| CHOP f | GGAAACAGAGTGGTCATTCCC |
| CHOP r | CTGCTTGAGCCGTTCATTCTC |
| atf4 f | ATGACCGAAATGAGCTTCCTG |
| atf4 r | GCTGGAGAACCCATGAGGT |
| Upp1 f | TGATTGCCCCGTCAGACTTT |
| Upp1 r | CACCAACGCACCTGATGAAG |
| gadd34 f | ATGATGGCATGTATGGTGAGC |
| gadd34 r | AACCTTGCAGTGTCCTTATCAG |
| β-actin f | AGA GGG AAA TCG TGC GTG AC |
| β-actin r | CAA TAG TGA TGA CCT GGC CGT |

(610832) from BD Biosciences (San Jose, CA), and LC3b (L8918) from Sigma (Taufkirchen, Germany).

Afterward, cells were washed three times for 5 min in PBS and incubated for 1 h with secondary antibodies labeled with Alexa fluorophores (1:1,000) and Alexa-488-phalloidin (1:200) from Thermo Fisher Scientific (Waltham, MA) at RT. Subsequently, cells were washed twice with PBS and stained with Dapi, and then mounted onto glass slides with 0.1 g/ml Mowiol.

For cholesterol staining, the Cholesterol Cell-Based Detection Assay Kit (Cayman Chemical Item No. 10009779) was used following the manufacturer's instructions.

**Confocal microscopy and analysis**

Immunofluorescent samples were analyzed using a Laser Scanning Confocal Microscope (Olympus Fluoview 1200, Olympus, Tokyo, Japan) equipped with an Olympus UPlanSApo 60× 1.35 and an UPlanSApo 40× 1.25Sil Oil immersion objective (Olympus, Tokyo, Japan) at a resolution of app. 100 µm/pixel (60×) and 600 nm step size. For the quantification of fluorescent intensity using Fiji software (ImageJ, v.2.0.0-rc-69/1.52p), individual images after background subtraction with a minimum of 20 cells were analyzed. Mean fluorescent signal was calculated per cell by using freehand selections. For the particle quantification of LC3-II, using the ImageJ plug-in, fluorescent dots with a pixel from 0 to 10 and circularity from 0.0 to 1.0 were included. The mean fluorescence per cellular area was calculated by dividing the overall intensity of particles with the cellular area in the same field.

analyzed in a MACSQuant 16 Analyzer. The Annexin-positive cells were quantified with a 488 nm laser (filter 525/50nm) and the PI-positive cells with a 488 nm laser (filter 615/20 nm). The percentage of apoptotic cells corresponds to the sum of two populations: Annexin V-positive /PI-negative cells (early apoptosis) and Annexin V-positive/PI-positive cells (late apoptosis). The gating strategy was established according to control cell populations.

**Immunofluorescence analysis**

Immunofluorescence was performed in cells grown on glass slides (Thermo Fisher FALC354108). HCT116 cells were fixed for 15 min in 4% paraformaldehyde, washed twice for 5 min in PBS, and permeabilized in 0.1 % Triton X-100 in PBS for 10 min at RT. After two more washes for 5 min in PBS, cells were blocked in 10% horse serum for 10 min at RT and subsequently treated with primary antibodies in 5% horse serum for 1 h. Primary antibodies used were Lamp1 (553792) and p62

### The paper explained

**Problem**
Drug development is a time-, cost-, and resource-consuming process with no guarantee of success, even with late-stage molecules. An elegant way to circumvent some of the before-mentioned obstacles is the repurposing of drugs outside their initially intended indication.

**Results**
In this study, we conducted a combinatorial drug screen with approved drugs to identify molecules which synergistically induce cell death in combination with niclosamide ethanolamine (NEN), a compound that induces metabolic stress in cancer cells. We identified specific antidepressants and a dopamine receptor antagonist to indeed synergistically trigger cancer cell death in combination with NEN, mediated by the induction of distinct cellular stress response pathways.

**Impact**
Importantly, the identified drug combination sensitized patient-derived pancreatic cancer organoids to chemotherapy and due to its excellent safety profile could potentially be included in standard tumor therapies in the future.

### ATP measurement

Cell were seeded in 96 wells and incubated overnight before treatment. The CellTiter Glo luminescence kit (Promega) was used according to the manufacturer's protocol to determine cellular ATP content.

## Data availability

Transcriptomic data (microarray) have been submitted to the GEO database at NCBI (GSE148682): https://www.ncbi.nlm.nih.gov/geo/query/acc.cgi?acc=GSE148682

**Expanded View** for this article is available online.

### Acknowledgements

This work was supported by an intramural drug development grant to G. H. and M. B. D., M. R., S. H., and M. B. D. are supported by the German Research Foundation (Deutsche Forschungsgemeinschaft, SFB1321 Modeling and Targeting Pancreatic Cancer, Project-ID 329628492). M. R. is supported by the German Cancer Aid Foundation (Max Eder Program, Deutsche Krebshilfe 111273, M. R.) and the German Research Foundation (RE 3723/4-1). This work was partially supported by the Helmholtz Alliance "Aging and Metabolic Programming, AMPro" (J. B.; S. H.). We thank Stefan Ambos, Eva Lederer, and Marcos Rios Garcia for technical help and discussions, and Manfred Roesner for advice on drug development. We thank Luke Harrison for his support with figure design. Open Access funding enabled and organized by Projekt DEAL.

### Author contributions

GH, YK, and BB performed the majority of experiments. F-FT, PM, and LM performed experiments. KS, SL, and GH performed the drug screening. GH and KS developed drug screening protocol. IR performed the drug screening data analysis. KH supervised drug screening and provided experimental advice. OP provided experimental advice on hit selection and validation. JMMK performed bioinformatics analysis of omics data. DL supervised bioinformatics analysis. GH, ZD, and AS performed organoid experiments. MR provided organoids and experimental advice. SS and AZ performed microscopic analysis and provided experimental advice on IF stainings and autophagy-related assays. JT and AA performed metabolomics analysis. JA supervised metabolomics analysis. MI performed transcriptome analysis. JB supervised transcriptome analysis. All authors analyzed and/or interpreted data. GH, SH, and MBD designed and directed research. GH, MBD, and SH wrote the manuscript.

### Conflict of interest

The authors declare that they have no conflict of interest.

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
