## [Review Process File · EMBO Molecular Medicine]

Combination therapies induce cancer cell death through the integrated stress response and disturbed pyrimidine metabolism

Mauricio Berriel Diaz, Götz Hartleben, Kenji Schorpp, Yun Kwon, Barbara Betz, Foivos Tsokanos, Zahra Dantes, Arlett Schäfer, Ina Rothenaigler, José Monroy Kuhn, Pauline Morigny, Lisa Mehr, Sean Lin, Susanne Seitz, Janina Tokarz, Anna Artati, Jerzy Adamski, Oliver Plettenburg, Dominik Lutter, Martin Irmeler, Johannes Beckers, Maximilian Reichert, Kamyar Hadian, Anja Zeigerer, and Stephan Herzig

DOI: [10.15252/emmm.202012461](https://doi.org/10.15252/emmm.202012461)

Corresponding author(s): [Mauricio Berriel Diaz \(mauricio.berrieldiaz@helmholtz-muenchen.de\)](mailto:mauricio.berrieldiaz@helmholtz-muenchen.de) , [Stephan Herzig \(stephan.herzig@helmholtz-muenchen.de\)](mailto:stephan.herzig@helmholtz-muenchen.de)

Review Timeline:

Submission Date:	4th Apr 20
Editorial Decision:	13th May 20
Revision Received:	26th Nov 20
Editorial Decision:	22nd Dec 20
Revision Received:	24th Jan 21
Accepted:	27th Jan 21

Editor: *Lise Roth*

Transaction Report:

13th May 2020

Dear Dr. Berriel Diaz,

Thank you for the submission of your manuscript to EMBO Molecular Medicine, and please accept my apologies for the delay in getting back to you in these exceptional circumstances. We have now received feedback from the three reviewers who agreed to evaluate your manuscript. As you will see from the reports below, the referees acknowledge the interest of the study and are overall supporting publication of your work pending appropriate major revisions.

Addressing the reviewers' concerns in full will be necessary for further considering the manuscript in our journal, with the exception of the *in vivo* validation suggested by referee #2. Indeed, we would welcome treatment studies *in vivo*, but these will not be required for publication of your manuscript.

Acceptance of the manuscript will entail a second round of review. EMBO Molecular Medicine encourages a single round of revision only and therefore, acceptance or rejection of the manuscript will depend on the completeness of your responses included in the next, final version of the manuscript. For this reason, and to save you from any frustrations in the end, I would strongly advise against returning an incomplete revision.

When submitting your revised manuscript, please carefully review the instructions that follow below. Failure to include requested items will delay the evaluation of your revision:

- 1) A .docx formatted version of the manuscript text (including legends for main figures, EV figures and tables). Please make sure that the changes are highlighted to be clearly visible.
- 2) Individual production quality figure files as .eps, .tif, .jpg (one file per figure).
- 3) A .docx formatted letter INCLUDING the reviewers' reports and your detailed point-by-point responses to their comments. As part of the EMBO Press transparent editorial process, the point-by-point response is part of the Review Process File (RPF), which will be published alongside your paper.
- 4) A complete author checklist, which you can download from our author guidelines (<https://www.embopress.org/page/journal/17574684/authorguide#submissionofrevisions>). Please insert information in the checklist that is also reflected in the manuscript. The completed author checklist will also be part of the RPF.
- 5) Please note that all corresponding authors are required to supply an ORCID ID for their name upon submission of a revised manuscript.
- 6) Before submitting your revision, primary datasets produced in this study need to be deposited in an appropriate public database (see

<https://www.embopress.org/page/journal/17574684/authorguide#dataavailability>).

The accession numbers and database should be listed in a formal "Data Availability " section (placed after Materials & Method). Please note that the Data Availability Section is restricted to new primary data that are part of this study.

7) We would also encourage you to include the source data for figure panels that show essential data. Numerical data should be provided as individual .xls or .csv files (including a tab describing the data). For blots or microscopy, uncropped images should be submitted (using a zip archive if multiple images need to be supplied for one panel). Additional information on source data and instruction on how to label the files are available at .

8) Our journal encourages inclusion of *data citations in the reference list* to directly cite datasets that were re-used and obtained from public databases. Data citations in the article text are distinct from normal bibliographical citations and should directly link to the database records from which the data can be accessed. In the main text, data citations are formatted as follows: "Data ref: Smith et al, 2001" or "Data ref: NCBI Sequence Read Archive PRJNA342805, 2017". In the Reference list, data citations must be labeled with "[DATASET]". A data reference must provide the database name, accession number/identifiers and a resolvable link to the landing page from which the data can be accessed at the end of the reference. Further instructions are available at .

9) We replaced Supplementary Information with Expanded View (EV) Figures and Tables that are collapsible/expandable online. A maximum of 5 EV Figures can be typeset. EV Figures should be cited as 'Figure EV1, Figure EV2' etc... in the text and their respective legends should be included in the main text after the legends of regular figures.

- Additional Tables/Datasets should be labeled and referred to as Table EV1, Dataset EV1, etc. Legends have to be provided in a separate tab in case of .xls files. Alternatively, the legend can be supplied as a separate text file (README) and zipped together with the Table/Dataset file. See detailed instructions here: .

10) The paper explained: EMBO Molecular Medicine articles are accompanied by a summary of the articles to emphasize the major findings in the paper and their medical implications for the non-specialist reader. Please provide a draft summary of your article highlighting

11) For more information: There is space at the end of each article to list relevant web links for further consultation by our readers. Could you identify some relevant ones and provide such information as well? Some examples are patient associations, relevant databases, OMIM/proteins/genes links, author's websites, etc...

12) Every published paper now includes a 'Synopsis' to further enhance discoverability. Synopses are displayed on the journal webpage and are freely accessible to all readers. They include a short stand first (maximum of 300 characters, including space) as well as 2-5 one-sentences bullet points that summarizes the paper. Please write the bullet points to summarize the key NEW findings. They should be designed to be complementary to the abstract - i.e. not repeat the same text. We encourage inclusion of key acronyms and quantitative information (maximum of 30 words / bullet point). Please use the passive voice. Please attach these in a separate file or send them by email, we will incorporate them accordingly.

Please also suggest a striking image or visual abstract to illustrate your article. If you do please provide a png file 550 px-wide x 400-px high.

13) As part of the EMBO Publications transparent editorial process initiative (see our Editorial at <http://embomolmed.embopress.org/content/2/9/329>), EMBO Molecular Medicine will publish online a Review Process File (RPF) to accompany accepted manuscripts.

In the event of acceptance, this file will be published in conjunction with your paper and will include the anonymous referee reports, your point-by-point response and all pertinent correspondence relating to the manuscript. Let us know whether you agree with the publication of the RPF and as here, if you want to remove or not any figures from it prior to publication.

I look forward to receiving your revised manuscript.

Yours sincerely,

Lise Roth

Lise Roth, PhD
Editor
EMBO Molecular Medicine

To submit your manuscript, please follow this link:

Link Not Available

Photos 400-800 DPI

*Additional important information regarding figures and illustrations can be found at <http://bit.ly/EMBOPressFigurePreparationGuideline>

***** Reviewer's comments *****

Referee #1 (Comments on Novelty/Model System for Author):

For similar experiments (same models, same treatments), the results are expressed according to very different scales and values (experiments of qPCR, cytotoxicity, caspase activity). It is difficult to interpret the results and determine if the variations observed are relevant, in particular because positive controls are missing.

Referee #1 (Remarks for Author):

In the present study, Hartlebel et al., evaluated a new combination of niclosamide and tricyclic antidepressants against colon cancer models. They described several molecular pathways involved in the cytotoxic process leading to lethal metabolic stress. The authors present an interesting study with relevant models (colon cancer cell lines and organoids derived from the patient). But the results are essentially descriptive and it is difficult to make the functional link between each molecular pathway described (IRS, catabolic CLEAR network, pyrimidic homeostasis, DNA damage). Each pathway is identified using mainly the qPCR approach of several genes and need to be confirmed by other approaches as proposed below. In addition, the study is not entirely well designed and several protocols need to be improved to confirm their results.

First, for many experiments, the authors used a viability and toxicity test based on the measurement of fluorescence and luminescence to assess the combination of drugs with effect. The results are expressed as a ratio between the RFU / RLU. But for each control condition, the ratio is different (eg: using the HCT116 cell line: Fig 1A (Ratio Control = 1), Fig 1D Ratio = 0.000005). In this condition, it is really difficult to assess the relevance of their results. The authors should

determine a more appropriate representation. From a monolayer culture, the authors could confirm the death and viability of the cells by flow cytometry using fluorescent probes (propidium iodide, calcein-AM, Sytox, ethidium bromide, etc.).

The induction of apoptosis was evaluated using a fluorimetric test based on the activity of caspase. But Authours did not include a positive control (like staurosporine) and a pan-caspase inhibitor (zVAD) to confirm the specificity of the measure. Experiments should be repeated with the appropriate control.

The authors did not include the validation of SiATF4 and siCHOP in the manuscript.

The integrated stress response has been shown primarily using the expression of CHOP mRNA. Western blot experiments could also be useful to fully describe the ISR molecular pathway, including the expression of Grp78, Phospho-eIF2a.

It is also really difficult to understand why the relative expression of the CHOP mRNA is between 0-15 in the left panel of Figure 2A and between 0-0.020 in the right panel of Figure 2A. According to legend, it is the same cell line and the same treatment condition. This is exactly the same problem for the DHODH gene (Fig 5). I recommend presenting the qPCR result as a fold change of control condition (by the $\Delta\Delta$ CT method of the pfall method)

Referee #2 (Remarks for Author):

General remarks: In this manuscript, Hartleben and colleagues explored a novel combination of anti-cancer therapeutics, with a particular focus on inducing metabolic vulnerabilities in a variety of cancer cell lines. Using high-throughput drug screening, they identified drug combinations that caused a starvation-like lethal catabolic response. By combining the mitochondrial uncoupler Niclosamide Ethanolamine (NEN) with dopamine receptor antagonist Domperidone or several tricyclic antidepressants (TCAs), the authors confirmed strong anti-cancer effects in both 2D and 3D cancer cell line cultures, as well as patient-derived pancreatic cancer organoids. While the search for novel, combinatorial therapeutic strategies aimed at hampering cancer metabolism is pressing and appreciated, the manuscript suffers from lack of detail and coherence. Essential aspects of the setup of several key experiments (including the HT-screen) are not sufficiently explained, and many experiments have been performed using only one cell line (HCT-116). Moreover, while representing an important aspect of the paper, the experiments involving patient-derived organoids are insufficiently presented. The following comments will thus need to be addressed:

General/major points:

1. Throughout the manuscript, the authors use the different cell lines in an inconsistent way (eg. Fig1 and EV1: 3D toxicity done in HCT-116 and U87 but not BxPC3, 2D caspase activity in HCT-116 and BxPC3 but not U87). Please consider performing all key experiments in all the three cell lines used (HCT-116, U87, BxPC3) to improve consistency and increase the impact of the results.
2. The drug screen would have been more meaningful if multiple concentrations and time points were used. Implementing this now may not be feasible, but please elaborate on how these specific settings were selected. Also, for each experiment please specify how many technical/biological replicates were used.
3. Most graphs generally lack information with regards to the experiment, for example which cell

lines were used and under which treatment conditions. This information has to be found in the figure legends or methods, but the results would be much easier to interpret when this information would be provided directly in the figure. The same applies to microscopy images - they should also contain clear color coding information.

4. It is well documented that considerable differences in cell interaction, drug sensitivity and metabolism exist between 2D and 3D cell cultures. The latter is being considered as a better predictive value for pre-clinical drug research. The authors should therefore carefully explain the rationale of using both models in their study.

5. Usage of multiple assays to confirm key findings is highly recommended to increase the overall reproducibility and the impact of the results. For example, please provide a proper cell death/apoptosis measurement (e.g. annexin V/propidium iodide, TUNEL, cleaved caspase 3 stainings) at least for the lead compounds.

6. Following on the previous point, to make a convincing point about the role of autophagy, the authors should better characterize the autophagy pathway and follow the accepted guidelines for the use and interpretation of autophagy (PMID26799652). For example, the use of dual sensor GFP-mCherry-LC3, next to qPCR data, additional immunoblots and their correct quantification for autophagy markers LC3 and p62 should be added, as well as the use of positive controls such as bafilomycin or chloroquine should be included. All of the above should facilitate better interpretation of the presented autophagy data.

7. IC50 dose-response curves of drugs would be helpful, at least for all lead compounds used in validation experiments.

8. The experimental conditions for the RNAseq are not clearly specified anywhere in the manuscript. A detailed description is needed to properly interpret the results.

9. A better integration of the RNA-seq and metabolomics data with the rest of the results is necessary. For example, the authors state induction of the CLEAR network upon combinatorial treatment, which should also be apparent from their transcriptomics data.

10. The representation of qPCR results is confusing. The scale of the "rel. expr. levels" on the y-axis is often highly different between panels, whereas the conditions are the same (the expression would be expected to be similar when the experimental conditions are the same). For example, in Fig. 2A and 3A.

11. In Figure 3A, for both TFE3 and CD68 expression, how is it possible that the effect of NEN treatment alone (2nd bar in all graphs) on relative expression levels is not the same between left and right panes (in the left panel it is significantly increased compared to the control, whereas in the right panel this is not the case)? Similar questions arise in Figure EV3A (NEN treatment alone, 2nd bar in left and right graphs) and in Figure 5D (control + A77 treatment in left and right panel have quite a different level of caspase activity). Also, although different cell lines, the researchers should comment on the differences in toxicity effects for control and ISRIB upon NEN+Domp treatment in Figure 2E (significant difference) and Figure EV2B (no difference). The latter is a clear example of why all experiments should be shown in all cell lines (see major comment 1).

12. Protein validation of key transcriptomic findings (qPCR or RNA-seq based) would help to obtain a more complete picture of the treatment effects on the different cancer cell lines.

13. The patient-derived organoids provide an interesting and clinically relevant addition to the paper, and a more in-depth validation of the mechanistic effects of the different compounds in the organoids would considerably strengthen the manuscript, especially as the organoids are the only model mentioned in the abstract. Upon reading the abstract, the reader is under the misconception that the majority of data is generated in organoids, however this is not the case.

14. A strong conclusion of the work is currently lacking. Did the authors identify a combinatorial treatment regimen that outperforms all others?

15. In vivo validation of the key combinatorial treatment(s) efficacy would add substantial impact to this work.

Minor points:

1. The current title is too broad. Please revise it to reflect the main message of the manuscript in a more specific manner.
2. An overview/schematic of the HT drug screen(s) would aid interpretation of the screen results. In Figure 1B, the number of different drugs that were tested, could be indicated in the pie chart. The authors should describe how the combination of the two HT screens was performed.
3. The figure legends need to be revised. A more detailed description of each figure panel is required to correctly interpret the results. Also, figure legends are lacking for all supplementary figures.
4. Please specify in all figure legends the number of replicates used for each experiment/graph. Also, please indicate the individual biological replicates as dots in the bar graphs.
5. All immunoblots need to be densitometrically quantified.
6. Scale bars are missing/too small in most of the images. Indication of the performed stainings is also missing (Fig 2C, 6A, 6B, EV3C).
7. Overall figure aesthetics can be improved. Figures are often not properly aligned. Usage of colors for different conditions is not consistent across bar graphs, group distribution for each condition is not consistent between graphs (e.g. Fig 4E) etc.
8. The knock-down efficiency of all siRNAs needs to be provided.
9. In Figure EV1, the mere showing of brightfield images of cell death in U87 and BXPC3 (should be BxPC3) is insufficient. Proper quantification of cell death in these cell lines should be added as done in Figure 1D.
10. The authors should provide a reference for the ISRIB and A771726 inhibitors.
11. In Figure 3C, include the quantification of LAMP1 staining.
12. In Figure 3D, it is unclear whether these are technical replicates (regions of interests?) or biological replicates. Also, instead of measuring LC3 intensity, the number of LC3 foci should be quantified. Also, this imaging and quantification does not represent an increase in the lipidated form of LC3 (LC3-II; as mentioned in the manuscript), as the staining does not discriminate between LC3-I and LC3-II. Only the immunoblot in Figure 3E represents an increase in LC3-II, however, densitometry and a clear labeling of LC3-I and LC3-II bands should be added. Moreover, a measurement of the ratio of LC3-II/LC3-I should also be included.
13. In Figure 4F&G, the "NEN only" control condition is missing.
14. The GO- and WikiPathway gene set enrichment results need to be presented differently. In their current form, it is hard to identify which pathway each dot refers to. These results could also be provided in the form of a supplementary table.
15. For all figures reporting cellular toxicity and caspase activity assays in the manuscript (Fig. 1A&D, etc.), a clear definition of "ratio tox/viability rfu/rlu" should be provided. Also, the abbreviations of "rfu" and "rlu" themselves should be indicated in the figure caption.
16. The reported statistical significance between groups need to be reevaluated for all figures. Some statistically significant differences are irrelevant (e.g. Fig. 2D) and some statements in the manuscript are not reflected by statistically significant differences in the figures (e.g. Fig 2B). In Fig. 7, the way to show the results of statistical tests should be improved for easier data interpretation.
17. In Suppl. Fig. 1B, judged from the labeling at the bottom of the heatmap, the number of replicates does not match between groups. For example, in the NEN treatment group, where are the replicates NEN_1 and NEN_4 (this reviewer wonders whether the researchers excluded certain replicates, but more importantly not even the same ones in each group)? The same applies to the "double treatment" group.
18. The labeling "co" to refer to control conditions could be replaced by "ctrl" to adopt a more conventional nomenclature and avoid confusion when referring to the use of combined drugs.
19. The graphical abstract in Figure EV5 and the manuscript mention "synergism", but this reviewer

is not convinced that the researchers truly prove synergism with the performed assays.
20. The title of Figure EV1 has a typo "uncoupling".

Referee #3 (Remarks for Author):

This study by Hartleben and colleagues, sets out on the premise (with good rationales) that combining metabolic inhibitors may make cancer cells more vulnerable to secondary therapies. On the back of this, they identify domperidone and various TCAs, as sensitizers to cell death in combination with mitochondrial uncoupler. Investigating the mechanism basis of this synergy they describe roles for autophagy, integrated stress response, cholesterol metabolism converging on nucleotide metabolism to make cells more sensitive to DNA-damage (supported in the last figure). Study is interesting and timely, some aspects of the model proposed by the authors are supported by the corresponding data. However, in my opinion, many of the findings here need further experimental support, to better demonstrate causality/elucidate mechanism. These points are detailed below:

- The demonstration that the ISR is being activated (and is relevant for cell death) needs to be better described, the data presented here are all consistent with activation of ER stress leading to apoptosis, given that CHOP but not ATF4 is relevant for the cell death observed upon combination treatment, ATF4 is also upregulated in the presence of ISRIB (Fig 2C), indicating that its upregulation is not dependent on ISR.
- There is the implication in Fig 3. that in response to drug treatment TFE3, MITF are regulating autophagy and this is regulating metabolic homeostasis but this is never tested, i.e. does knockdown of either block the increase in autophagy observed and does inhibiting autophagy disrupt metabolic homeostasis (presumably increasing cell death) ?
- Gamma H2AX is used as a read out of DNA-damage that the authors conclude initiates cell death, however, DNA-damage occurs during cell death (as CAD cleaves DNA), whether the DNA-damage they observe is cause or consequence should be defined (for instance using caspase-inhibitors)
- Given the translational lean of this paper (and of course of the journal), its important to investigate the potential toxicity of these drug treatments in normal cells/healthy tissue as far as is possible.
- Throughout the ms. cell viability/caspase activity is provided in relative levels, for key expts. its important to note the actual level (%) cell death, since, for instance, a relative increase of 3 fold caspase activity could equate to a 1 to 3% increase in cell death, clearly this would be irrelevant when considering translating these findings.

Rebuttal to the reviewers

We thank the reviewers for their overall support and constructive criticism. The reviewers raised a number of very valid points that we have addressed by providing new data as well as further explanations and clarifications to the previously provided data. In our view, this resulted in a clearly improved revised version of the study, which was enabled by the valuable and specific advices provided by the other reviewers. Please find below the responses to the specific points. We have now addressed their comments on a point-by-point basis, particularly addressing common issues raised by several reviewers, i.e. an additional FACS-based analysis of cell death, protein level determination, normalization of qPCR results and a new validation of autophagy-related analyses. Please find our detailed responses below.

Referee #1 (Comments on Novelty/Model System for Author):

*For similar experiments (same models, same treatments), the results are expressed according to **very different scales and values** (experiments of qPCR, cytotoxicity, caspase activity). It is difficult to interpret the results and determine if the variations observed are relevant, in particular because positive controls are missing.*

We thank the reviewer for the careful evaluation of our study. As you will see below, we now provide additional data, including an additional readout for cell death effects (% apoptotic determined by flow cytometry-based apoptosis assays (including staurosporin as positive control)). Also, we re-evaluated previously shown data to provide a more uniform representation of the effects. We believe, that this enables a better assessment of the relevance of the findings and thank the reviewer for the specific advices that clearly improved the overall quality of the study.

Referee #1 (Remarks for Author):

*In the present study, Hartlebel et al., evaluated a new combination of niclosamide and tricyclic antidepressants against colon cancer models. They described several molecular pathways involved in the cytotoxic process leading to lethal metabolic stress. The authors present an interesting study with relevant models (colon cancer cell lines and organoids derived from the patient). But the results are **essentially descriptive and it is difficult to make the functional link between each molecular pathway described (IRS, catabolic CLEAR network, pyrimidic homeostasis, DNA damage)**. Each pathway is identified using mainly the qPCR approach of several genes and **need to be confirmed by other approaches as proposed below**. In addition, the study is not entirely well designed and **several protocols need to be improved to confirm their results**.*

1. First, for many experiments, the authors used a viability and toxicity test based on the

measurement of fluorescence and luminescence to assess the combination of drugs with effect. The results are expressed as a ratio between the RFU / RLU. But for each control condition, the ratio is different (eq: using the HCT116 cell line: Fig 1A (Ratio Control = 1), Fig 1D Ratio = 0.000005. In this condition, it is really difficult to assess the relevance of their results. The authors should determine a more appropriate representation. From a monolayer culture, the authors could confirm the death and viability of the cells by flow cytometry using fluorescent probes (propidium iodide, calcein-AM, Sytox, ethidium bromide, etc.).

As outlined in more detail in our response to the reviewer's point 2, we have confirmed the cell death inducing effects of drug treatments using flow cytometry-based assays. However, please note that different assays have been used for the assessment of viability depending on the culturing method, presenting the ratio of RFU/RLU for cells cultured in 3D (Promega CellToXGreen and RealTime-Glo™ MT Cell Viability kits) and RFU/RFU for the assays performed in 2D (MultiTox-Fluor Multiplex Cytotoxicity Assay), as well as caspase activity assays (Apo One Homogenous Caspase-3/7 Assay). The corresponding labelling has been improved for clarification. The difference of the ratios in Figure 1A vs 1D stems from the different setup in experiments and drugs used in the experiments (2DG treatment versus TCA treatment). Differences in cell numbers due to counting variations can also contribute to some extent to ratio differences in inter-experiments comparisons. The example given by the reviewer is however the only major numerical difference in comparison to the remaining results, where exclusively NEN+Domperidone/TCA were used as drug treatments (when comparing the values for identical assays) and did not perform further normalization of the original values (as we did for qPCR analyses, see below).

*2. The induction of apoptosis was evaluated using a fluorimetric test based on the activity of caspase. But Authors did not include a **positive control (like staurosporine) and a pan-caspase inhibitor (zVAD)** to confirm the specificity of the measure. Experiments should be repeated with the appropriate control.*

The reviewer raised the very valid points. Also, please note that the additional readout confirming the relevance cell death effects upon combined drug treatments has been also requested by the other two reviewers (Reviewer #5, point 5 and Reviewer #3 point 5).

As requested, we have now performed FACS-based analyses to assess cell death effects for single and combined drug treatments. We detected Annexin V- and PI-positive cells in order to determine the percentage of apoptotic cells along with staurosporin treatment as positive control as well as concomitant treatment with the pan-caspase inhibitor zVAD (for NEN+Domperidone combined treatments), the latter in order to counteract the induction of apoptosis under combined treatment conditions. The results are shown in the new Figures 1E (for HCT116 and U87 cells upon NEN/Domperidone single and combined treatments) and in Figures EV1E (for HCT116 and U87 cells for the combined treatments with NEN, Domperidone and the 4 tricyclic antidepressant (TCA) drugs).

The treatment conditions (drug concentrations and treatment durations) in the FACS analyses were identical to the conditions used for the respective toxicity or caspase assays shown previously. Under these conditions and depending on the cell line as well as the specific drug combinations, we observed up to 70%-80% apoptotic cells (combining Annexin V-positive cells in early apoptosis and AnnexinV/PI double-positive cells). In some approaches the proportion of apoptotic cells were lower (around 30%). However, please note that the analyses refer to fixed time points and the kinetics of cell death induction vary depending on the cell line as well as the different drug combinations. Upon treatment prologation, combined treatments will result in a virtually complete death of cancer cells under investigation (in contrast to the treatment of untransformed cells (primary hepatocytes), now included in new Figure 1F upon request of reviewer #3). The induction of apoptosis upon combined drug treatment were partly but significantly rescued by co-treating the cells with the caspase inhibitor zVAD. Additionally, we included samples treated in parallel with Staurosporin as a positive control for apoptosis induction. For each cell line, the length of the Staurosporin treatment (100 μ M) matched that of the other conditions (i.e 24 hours for the HCT116 cells and 48 hours for the U87) and resulted in comparable induction of cell death. We also tried respective analyses with the pancreatic cancer cell line BxPC3. However, the FACS-based analysis was substantially hampered by the fact that these cells are very adherent and the required intense trypsinisation was not compatible with an appropriate conservation of cell integrity for the subsequent FACS analysis (which is generally more difficult with adherent cells). Taken together, the new analyses confirmed the relevance of cytotoxic effects of the combined drug treatments in HCT116 as well as U87 cells.

3. The authors did not include the validation of siATF4 and siCHOP in the manuscript.

We apologize for this and have now included the validation of the knockdown efficiency of siATF4 and siCHOP in the new Figure EV1D. Both siRNA are remarkably efficient in knocking down the respective target genes.

4. The integrated stress response has been shown primarily using the expression of CHOP mRNA. Western blot experiments could also be useful to fully describe the ISR molecular pathway, including the expression of Grp78, Phospho-eIF2a.

For addressing the reviewer's request, we have now performed extensive Western blot analyses for CHOP, ATF4 as well as total and phosphorylated eIF2a, including all three cell lines and all drug combinations as well as respective quantifications. Due to space limitations we could not allocate all the data in main and EV Figures and therefore show selected analyses representing the different cell lines as well as Domp and one representative TCA (Amitryptiline)(new Figures 2B and Figures EV2B, EV2C (please note that Figure EV2C shows the corresponding quantifications of WBs shown the Figures 2B and EV2B, as also indicated in the Figure). However, the remaining analyses are provided as Appendix Figure S4A and S4B.

In agreement with the qPCR data, the quantified Western blot analyses overall confirmed the induction of the ISR on protein level (represented by CHOP, ATF4 and p-/t-eIF2a), showing occasionally milder effects of single NEN treatment and a clearly more pronounced and consistent induction under combined drug treatment conditions.

5. It is also really difficult to understand why the relative expression of the CHOP mRNA is between 0-15 in the left panel of Figure 2A and between 0-0.020 in the right panel of Figure 2A. According to legend, it is the same cell line and the same treatment condition. This is exactly the same problem for the DHODH gene (Fig 5). I recommend presenting the qPCR result as a fold change of control condition (by the deltadeltaCT method of the pfafl method)

We apologize for this inconsistency that was based on the expression levels of the house-keeping gene for the normalization of the ct-values of the genes of interest, which can differ in inter-experimental conditions and upon Domperidone or TCA drug treatment. However, we of course confirmed that the expression of the respective house-keeping genes were stable in intra-experimental comparisons. In order to show fold-change differences relative to the respective control condition, we re-calculated the qPCR results as suggested by the reviewer for better comparison. All qPCR graphs have been exchanged accordingly, now validating our initial analyses.

Referee #2 (Remarks for Author):

*General remarks: In this manuscript, Hartleben and colleagues explored a novel combination of anti-cancer therapeutics, with a particular focus on inducing metabolic vulnerabilities in a variety of cancer cell lines. Using high-throughput drug screening, they identified drug combinations that caused a starvation-like lethal catabolic response. By combining the mitochondrial uncoupler Niclosamide Ethanolamine (NEN) with dopamine receptor antagonist Domperidone or several tricyclic antidepressants (TCAs), the authors confirmed strong anti-cancer effects in both 2D and 3D cancer cell line cultures, as well as patient-derived pancreatic cancer organoids. While the search for novel, combinatorial therapeutic strategies aimed at hampering cancer metabolism is pressing and appreciated, **the manuscript suffers from lack of detail and coherence**. Essential aspects of the setup of **several key experiments (including the HT-screen) are not sufficiently explained**, and many experiments have been performed **using only one cell line (HCT-116)**. Moreover, while representing an important aspect of the paper, **the experiments involving patient-derived organoids are insufficiently presented**. The following comments will thus need to be addressed:*

We thank the reviewer for appreciating the general relevance of our findings and for the thorough evaluation of our study as well as the valuable advices, which resulted in an improved revised version of the manuscript. Please find below the responses to the specific points.

General/major points:

1. Throughout the manuscript, the authors use the different cell lines in an inconsistent way (eg. Fig1 and EV1: 3D toxicity done in HCT-116 and U87 but not BxPC3, 2D caspase activity in HCT-116 and BxPC3 but not U87). Please consider performing all key experiments in all the three cell lines used (HCT-116, U87, BxPC3) to improve consistency and increase the impact of the results.

We apologize for giving the impression that the different cell lines have been used in an arbitrary manner and appreciate that not all the key findings have been shown for all the three cell lines using both culturing methods (2D/3D). However, we would like to strongly emphasize that this was not due to inconsistencies in the results (and the application of an unacceptable “cherry picking strategy”), but partly due to technical limitations (as specified below) as well as limitations in the Figure space. However, we think that with the addition of new and the partly re-organization of previously shown data, the revised version of the manuscript has clearly improved with respect to a more consistent representation of the different experimental systems (as outlined specifically below). Nevertheless, HCT116 cells still represent the “main model system” for proof-of-principle of drug treatment effects and the underlying mechanisms and has therefore been used preferentially in the main Figures, selectively complemented by specific analyses in the other 2 cell lines (preferentially represented in respective EV Figures). With respect to your specific examples: We have now

performed the Toxicity assay also in BxPC3 cells cultured as spheroids, which is technically challenging, given that these cells are affected by the 3D culturing conditions conferring increased sensitivity to any kind of stress/treatment. Consistently, the toxicity data for all three cell lines in 3D, including the treatments with all drug combinations are now shown in the new Figure 1D. The caspase activity assays for HCT116 and BxPC3 are now shown together in the new Figure EV1D. Concerning the absence of the caspase assay for U87, please note that these cells are very treatment resistant in 2D culture, which is also illustrated in by the relatively lower efficacy of the treatment with staurosporine, used as positive control in the newly added FACS-based apoptosis assays (new Figures 1E and EV1E, for HCT116 and U87).

However, we think that with the addition of new data in the revised versions, we have generally improved the consistency concerning the representation of the different cell lines under investigation.

2. The drug screen would have been more meaningful if multiple concentrations and time points were used. Implementing this now may not be feasible, but please elaborate on how these specific settings were selected. Also, for each experiment please specify how many technical/biological replicates were used.

Before conducting drug screening, we determined the optimal concentrations for NEN (now included in Appendix Figure S1) and the positive control in several assay development experiments. We have hereby defined 1.2 μM NEN as the optimal concentration, since the cells only show minor increased toxicity at this concentration, but are sensitized to the positive control. The small molecules of the Prestwick and GPCR library were screened at $\sim 10\mu\text{M}$ in order to keep the concentration of the solvent DMSO at a maximum of 1%. In addition, a concentration of $10\mu\text{M}$ is an accepted standard approach for primary screening campaigns (PMID: 21091654). From experience with other screening projects, we know that the substances of the Prestwick and GPCR library show no or only mild toxic effects within a period of 24 hours at a concentration of $10\mu\text{M}$ and was therefore chosen as appropriate condition for the detection of synergistic Toxicity effects in combination with NEN. This explanations on the screening strategy have been added to the respective description of screening in the Experimental procedure section.

3. Most graphs generally lack information with regards to the experiment, for example which cell lines were used and under which treatment conditions. This information has to be found in the figure legends or methods, but the results would be much easier to interpret when this information would be provided directly in the figure. The same applies to microscopy images - they should also contain clear color coding information.

We have now systematically included more information directly into the Figure in order to facilitate the interpretation of the results.

4. It is well documented that considerable differences in cell interaction, drug sensitivity and metabolism exist between 2D and 3D cell cultures. The latter is being considered as a better predictive value for pre-clinical drug research. The authors should therefore carefully explain the rationale of using both models in their study.

We agree with the reviewer concerning the predictive value of 3D cultures and have accordingly made some changes concerning the allocation of the data. For instance, Figure 1D now shows the toxicity assays upon treatment with all drug combinations in all three cell lines. Corresponding assays showing induction of caspase 3/7 (for HCT116 and BxPC3) are now shown in the expanded view Figure EV1D. Generally, we think that providing data produced using different culturing conditions further confirms the relevance/robustness of the effects of the proposed drug combinations. Also, it enables the use of different assays (toxicity/viability assays; caspase activation assays), which in part require specific culturing conditions and have now been even extended upon request by the 3 reviewers by including FACS-based assessment of apoptosis (for which the cells have been treated in 2D culture). In the course of the study, we conducted a step-wise drug validation strategy starting from 2D screening and validation in 2D experiments (including different read-outs for cell death). From there, we escalated the study to 3D cell line spheroids as a model more closely resembling an in-vivo situation and finally validated our findings in patient derived organoids to address cellular heterogeneity of a tumor.

5. Usage of multiple assays to confirm key findings is highly recommended to increase the overall reproducibility and the impact of the results. For example, please provide a proper cell death/apoptosis measurement (e.g. annexin V/propidium iodide, TUNEL, cleaved caspase 3 stainings) at least for the lead compounds.

We thank the reviewer for this very valid point. Also, please note that additional readout confirming the relevance cell death effects upon combined drug treatments has been similarly requested by the other two reviewers (Reviewer #1, point 1 and Reviewer #3 point 5). We therefore repeated the respective combined response (please see below).

As requested, we have now performed FACS-based analyses to assess cell death effects for single and combined drug treatments. We detected Annexin V- and PI-positive cells in order to determine the percentage of apoptotic cells along with staurosporin treatment as positive control as well as concomitant treatment with the pan-caspase inhibitor zVAD (for NEN+Domperidone combined treatments), the latter in order to counteract the induction of apoptosis under combined treatment conditions. The results are shown in the new Figures 1E (for HCT116 and U87 cells upon NEN/Domperidone single and combined treatments) and in Figures EV1E (for HCT116 and U87 cells for the combined treatments with NEN, Domperidone and the 4 tricyclic antidepressant (TCA) drugs).

The treatment conditions (drug concentrations and treatment durations) in the FACS analyses were identical to the conditions used for the respective toxicity or caspase assays

shown previously. Under these conditions and depending on the cell line as well as the specific drug combinations, we observed up to 70%-80% apoptotic cells (combining Annexin V-positive cells in early apoptosis and AnnexinV/PI double-positive cells). In some approaches the proportion of apoptotic cells were lower (around 30%). However, please note that the analyses refer to fixed time point and the kinetics of cell death induction vary depending on the cell line as well as the different drug combinations. Upon treatment prologation, combined treatments will result in all virtually complete death of cancer cells under investigation (in contrast to the treatment of untransformed cells (primary hepatocytes), now included in new Figure 1F upon request of reviewer #3). The induction of apoptosis upon combined drug treatment were partly but significantly rescued by co-treating the cells with the caspase inhibitor zVAD. Additionally, we included samples treated in parallel with Staurosporin as a positive control for apoptosis induction. For each cell line, the length of the Staurosporin (100 μ M) treatment matched that of the other conditions (i.e 24 hours for the HCT116 cells and 48 hours for the U87) and resulted in comparable induction of cell death. We also tried respective analyses with the pancreatic cancer cell line BxPC3. However, the FACS-based analysis was substantially hampered by the fact that these cells are very adherent and the required intense trypsinisation was not compatible with an appropriate conservation of cell integrity for the subsequent FACS analysis (which is generally more difficult with adherent cells). Taken together, the new analyses confirm the relevance of cytotoxic effects of the combined drug treatments in HCT116 as well as U87 cells.

6. Following on the previous point, to make a convincing point about the role of autophagy, the authors should better characterize the autophagy pathway and follow the accepted guidelines for the use and interpretation of autophagy (PMID26799652). For example, the use of dual sensor GFP-mCherry-LC3, next to qPCR data, additional immunoblots and their correct quantification for autophagy markers LC3 and p62 should be added, as well as the use of positive controls such as bafilomycin or chloroquine should be included. All of the above should facilitate better interpretation of the presented autophagy data.

We thank the reviewer for this very valuable advice which indeed enabled a more accurate analysis of autophagy and interpretation of the data. We now provide a state-of-the-art assessment of autophagy for individual and combined drug treatments, including quantified immunofluorescence (IF) staining for Lamp 1, LC3-II foci and p62 (new Figure 3B), as well as quantified immunoblots for LC3-II/I ratio and p62 (new Figures 3C and EV3C). As requested by the reviewer, we also determined autophagic flux as the ratio of LC3-II protein levels in the absence or presence of chloroquine treatment under different drug treatment conditions (new Figure 3D).

We confirmed the synergistic induction of autophagosome formation upon combined drug treatments (for single NEN, single Domp (IF), NEN+ Domp, NEN+Imiprimine (WB) or NEN+Amipramine (WB)), indicated by the marked induction of LC3-II (new Figures 3B, 3C and EV3C). Interestingly, p62 levels showed a very similar pattern of regulation, indicating a blockage of autophagosome clearance at the level of lysosomal degradation, which was in

agreement with the observed increase in the levels of the lysosomal marker Lamp1 (new Figures 3B, 3C and EV3C). Indeed, determination of autophagic flux by blocking lysosomal autophagosome clearance using chloroquine confirmed that autophagic flux was reduced under combined drug treatment conditions despite induction of autophagosome formation (new Figure 3D). It will be interesting to elucidate the exact molecular mechanism of autophagy blockage in the given context, which is however, beyond the scope of the present manuscript.

7. IC50 dose-response curves of drugs would be helpful, at least for all lead compounds used in validation experiments.

We performed dose-response analysis for all NEN as well as all hit compounds determined in the screens. This data has been included in Appendix Figure S1.

8. The experimental conditions for the RNAseq are not clearly specified anywhere in the manuscript. A detailed description is needed to properly interpret the results.

We apologize for not providing detailed information concerning the treatment conditions used in the transcriptomics analysis, which was actually generated using microarrays. This information is now provided along in the respective section in Materials and Methods. The experiment has been performed in HCT116 using the usual drug concentrations as for respective experiments throughout the manuscript: Vehicle control, 1.2 μ M NEN, 30 μ M Domperidone, and NEN+Domperidone. Cells were harvested after 16h, before the occurrence of major cell death observed in the toxicity/apoptosis assays after 24h. 3 out of 5 replicates were chosen for the microarray analysis based on RNA quality after isolation.

9. A better integration of the RNA-seq and metabolomics data with the rest of the results is necessary. For example, the authors state induction of the CLEAR network upon combinatorial treatment, which should also be apparent from their transcriptomics data.

In Appendix Figure S6, we have now included a heat map depicting the regulation of genes defining the CLEAR network as reported by Palmieri et al., 2011 (PMID: 21752829) upon combined NEN+Domp versus Control in HCT116 cells.

10. The representation of qPCR results is confusing. The scale of the "rel. expr. levels" on the y-axis is often highly different between panels, whereas the conditions are the same (the expression would be expected to be similar when the experimental conditions are the same). For example, in Fig. 2A and 3A.

We apologize for this inconsistency that was based on the expression levels of the house-keeping gene for the normalization of the ct-values of the genes of interest, which can differ inter-experimentally due to different conditions and upon Domperidone or TCA drug treatment. However, we of course confirmed that the expression of the respective house-

keeping genes were stable in intra-experimental comparisons. In order to show fold-change differences relative to the respective control condition, we re-calculated the qPCR results as suggested by the reviewer for better comparison. All qPCR graphs have been exchanged accordingly.

11. In Figure 3A, for both TFE3 and CD68 expression, how is it possible that the effect of NEN treatment alone (2nd bar in all graphs) on relative expression levels is not the same between left and right panes (in the left panel it is significantly increased compared to the control, whereas in the right panel this is not the case)? Similar questions arise in Figure EV3A (NEN treatment alone, 2nd bar in left and right graphs) and in Figure 5D (control + A77 treatment in left and right panel have quite a different level of caspase activity). Also, although different cell lines, the researchers should comment on the differences in toxicity effects for control and ISRIB upon NEN+Domp treatment in Figure 2E (significant difference) and Figure EV2B (no difference). The latter is a clear example of why all experiments should be shown in all cell lines (see major comment 1).

We acknowledge that NEN treatment alone was occasionally inducing the respective readouts in a variable manner between different experiments. However, these responses were generally clearly lower than the effects of combined treatments.

MBD: Fig EV3A, left and right panel, MBD: I did never really get the point of this figure....?

With respect to Figure 5D, A77 treatment, left versus right panel: This is presumably due to combined inter-assay and inter-experiment variability. When running the caspase assay, minor differences in incubation times can lead to differences in absolute numbers (the longer the incubation the higher the activity). To exactly time every experiment to achieve similar absolute numbers might be impossible. However, it is the fold changes within one experiment that matter for the treatment effect.

Although the reviewer is of course right that this represents a certain level of inconsistency at the lower level of control conditions, we think that this does not question the relevance of the finding concerning the substantial sensitization of combined drug treatments (in this case at sub-lethal levels) to DHODH inhibition by A77 treatment.

Concerning the difference between Figure 2E (HCT116 in 3D, now Figure 2F in the revised version) and Figure EV2B (BxPC3 in EV2D, now Figure EV2E in the revised version): For unknown reasons, the induction of toxicity upon Domp+NEN treatment in BxPC3 cells (now Figure EV2E) did obviously not work in this specific approach, and therefore no rescue by co-treatment with ISRIB could be observed (in contrast to the other 4 drug combinations, which show comparable effects, also as compared to HCT116 in the new Figure 2F). In response to the reviewer's major point 1, we have now included the toxicity assay for BxPC3 cells in 3D (included in the new Figure 1D), confirming the response of BxPC3 cells to NEN+Domp combined treatments, which further suggests that lack of effect in Figure EV2F is probably due to technical reasons.

12. Protein validation of key transcriptomic findings (qPCR or RNA-seq based) would help to obtain a more complete picture of the treatment effects on the different cancer cell lines.

A similar point has been raised by reviewer #1 (point 4) with respect to the validation of the induction of ISR genes on protein level. We have therefore performed extensive Western blot analyses of ISR marker genes CHOP, ATF4 and p-/t-eIF2a as well as for UPP1 for all three cell lines and virtually all drug combinations as well as respective quantifications. The results for the ISR genes are shown in the new Figures 2B and Figures EV2B, EV2C as well as Appendix Figures S4A and S4B. Western blot analyses including densitometric quantifications for UPP1 are shown in the new Figures EV4B and EV4C as well as Appendix Figure S8. The previously provided Western blots showing nuclear translocation of TFE3 and MITF upon NEN+Domp double treatment have been moved Figure EV3B. The new Western blot analyses overall confirmed that the induction of key genes previously determined on mRNA is also reflected on protein levels.

13. The patient-derived organoids provide an interesting and clinically relevant addition to the paper, and a more in-depth validation of the mechanistic effects of the different compounds in the organoids would considerably strengthen the manuscript, especially as the organoids are the only model mentioned in the abstract. Upon reading the abstract, the reader is under the misconception that the majority of data is generated in organoids, however this is not the case.

This is a justified criticism. The analysis of patient-derived organoids have been performed in collaboration with partners in the clinics. Therefore, we do not have unlimited access to this model systems, which are also not suitable for all kinds of different analyses and/or would require extensive optimizations which unfortunately could not be realized. However, as some mechanistic insight, we now provide gene expression analyses for a number of target genes defined in this study (CHOP, UPP1), which are shown in the new Figure EV5C. In the abstract we have removed one statement referring to organoids and now only mention the sensitization of patient-derived organoids to standard chemotherapy with Paclitaxel by the different drug combinations from this study.

14. A strong conclusion of the work is currently lacking. Did the authors identify a combinatorial treatment regimen that outperforms all others?

At this point, we can for sure not claim that our combinatorial treatment regimen outperforms all others, which would be a rather bold statement per se, given that in many cases these kinds of conclusions can only be reached after extensive clinical testing. We believe that the strength and relevance of our findings resides in the fact that we identified drug combinations that (i) markedly boost the cytotoxic effects of some drugs that have been or are currently in clinical testing against cancer (Niclosamide in combination with Enzalutamide against prostate cancer, NCT02532114; Desipramine against small cell

lung cancer and high-grade neuroendocrine Tumors, NCT01719861; Imipramine against triple-negative breast cancer, NCT03122444), and that (ii) we provide novel insights into central components of the molecular mechanisms contributing to induced anti-cancer efficacy upon combined treatments.

The latter point contributes to the general understanding of the response of cancer cells to metabolic stress conditions and opens the possibility to identify novel targets as part of the affected pathways.

Also, given that the efficacy of the tested drugs in clinical settings has not been yet confirmed or the respective studies have been terminated due to the lack of efficacy, our findings offer a promising novel strategy to augment the efficacy by combinatorial treatment regimen, thereby facilitating the repurposing of TCAs for cancer therapy in a potential broad efficacy spectrum. Notably, with the exception of Domperidone, all drugs have favorable safety profiles and are well tolerated over prolonged periods. Interestingly, we did not observe any considerable cytotoxicity in untransformed cells (at least in primary hepatocytes), which is in line with our initial idea of identifying metabolic vulnerabilities specific for cancer cells.

15. In vivo validation of the key combinatorial treatment(s) efficacy would add substantial impact to this work.

This is indeed a valid point and the subject of follow-up studies that we intend to pursue in the future. Along this line, we are currently working on different approaches for targeted delivery of the mitochondrial uncoupler component of combined treatments into the tumor in vivo. We believe that improving the pharmacokinetic characteristics of NEN (and potential additional mitochondrial uncouplers) by these means is required to develop a combinatorial treatment regimen with the potential to outperform some of the established chemo- or targeted therapies and pave the way to subsequent clinical translation of the proposed treatment concept. However, we believe that these ongoing mid- to long-term efforts are beyond the scope of the present study (please consider in this context also the response to your previous point concerning the overall relevance).

Minor points:

1. The current title is too broad. Please revise it to reflect the main message of the manuscript in a more specific manner.

The title of the study has been changed to “Pharmacological combination therapies induce cancer cell death through the an integrated stress response and disturbed pyrimidine metabolism”.

2. An overview/schematic of the HT drug screen(s) would aid interpretation of the screen results. In Figure 1B, the number of different drugs that were tested, could be indicated in the pie chart. The authors should describe how the combination of the two HT screens was performed.

The total number of tested drugs in the primary screen (1280) has been added to figure 1B. Additional information to the general screening strategy has been included to the respective methods section.

3. The figure legends need to be revised. A more detailed description of each figure panel is required to correctly interpret the results. Also, figure legends are lacking for all supplementary figures.

Figure legends have been revised and additional information to individual figure panels have been added. Figure legends have been included directly in the Appendix Figures (previously Supplemental Figure legends).

4. Please specify in all figure legends the number of replicates used for each experiment/graph. Also, please indicate the individual biological replicates as dots in the bar graphs.

N-numbers have been included in the Figure legends.

5. All immunoblots need to be densitometrically quantified.

We now provide densitometric quantifications for most immunoblots, particularly for those that include 3 replicates per condition. Due to space limitations we did not include the quantifications of more “qualitative” immunoblot analyses including individual samples for different conditions and obvious (black and white) effects.

6. Scale bars are missing/too small in most of the images. Indication of the performed stainings is also missing (Fig 2C, 6A, 6B, EV3C).

For addressing this reviewer's major point 6, we provided new IF images for autophagy-related markers (Lamp1, LC3-II foci, p62) in the new Figure 3, including appropriate labelling and visible scale bars (please note that Fig. 2C did not include any images and we therefore assumed that the reviewer was referring to the former Fig. 3C). Figures 6A and 6B have been corrected accordingly.

7. Overall figure aesthetics can be improved. Figures are often not properly aligned. Usage of colors for different conditions is not consistent across bar graphs, group distribution for each condition is not consistent between graphs (e.g. Fig 4E) etc.

We have now improved overall Figure aesthetics including a consistent color coding across bar graphs (whenever applicable).

8. The knock-down efficiency of all siRNAs needs to be provided.

The knock-down efficiencies of siChop and siATF4 have now been added to Figure EV2

9. *In Figure EV1, the mere showing of brightfield images of cell death in U87 and BXP3 (should be BxPC3) is insufficient. Proper quantification of cell death in these cell lines should be added as done in Figure 1D.*

We have removed the bright field images and included the assessed cell death upon the respective treatments in U87 and BxPC3 according to Figure 1D. Please see new Figures EV1A.

10. *The authors should provide a reference for the ISRIB and A771726 inhibitors.*

The respective references have been included into the manuscript, for ISRIB (10.1074/jbc.270.38.22467) and A771726 (10.7554/eLife.00498).

11. *In Figure 3C, include the quantification of LAMP1 staining.*

As described in the response to your point 6, we now provide new autophagy-related data in Figure 3, including IF staining for LAMP1, as well as the respective quantifications of the IF staining, as requested (see new Figure 3B). Please note that the previous Figure 3C showing Lamp1 staining has been replaced.

12. *In Figure 3D, it is unclear whether these are technical replicates (regions of interests?) or biological replicates. Also, instead of measuring LC3 intensity, the number of LC3 foci should be quantified. Also, this imaging and quantification does not represent an increase in the lipidated form of LC3 (LC3-II; as mentioned in the manuscript), as the staining does not discriminate between LC3-I and LC3-II. Only the immunoblot in Figure 3E represents an increase in LC3-II, however, densitometry and a clear labeling of LC3-I and LC3-II bands should be added. Moreover, a measurement of the **ratio of LC3-II/LC3-I** should also be included.*

Given the renewed assessment of autophagy, the former Figure 3D has been removed. The new Figure 3B includes IF staining and quantification for LC3 foci (representing LC3-II), p62 and Lamp1. The respective quantification methods have been described in the Experimental Procedure section. As advised by this reviewer, we determined the LC3-II/I ratios from the immunoblots shown in the new Figures 3C and EV3B, showing a marked induction LC3-II/I ratio indicative of autophagosome formation/accumulation under combined treatment conditions.

13. *In Figure 4F&G, the "NEN only" control condition is missing.*

Since the experiment aims to address to describe events which take place under apoptosis inducing conditions, we only included the combinatorial treatment (single treatment does not lead to major apoptosis). In particular in 4F, the UPP1 knockdown aims to show the “rescue” effect and does not aim to address the contribution of the singular drugs. Therefore, we consider the vehicle-treatment control as the appropriate one.

14. The GO- and WikiPathway gene set enrichment results need to be presented differently. In their current form, it is hard to identify which pathway each dot refers to. These results could also be provided in the form of a supplementary table.

The presentation of the Wikipathway GSEA (now Appendix Figure S5) has been exchanged for a version facilitating the assignment of individual dots to the respective pathways. The GO-GSEA include lines connecting the pathways to the respective dots (now Appendix Figure S2). We hope that the representation is now sufficiently improved.

15. For all figures reporting cellular toxicity and caspase activity assays in the manuscript (Fig. 1A&D, etc.), a clear definition of "ratio tox/viability rfu/rlu" should be provided. Also, the abbreviations of "rfu" and "rlu" themselves should be indicated in the figure caption.

We have now included this information into the Material and Methods section referring to the respective assays and the manufacturers. We hope that this is acceptable to this reviewer.

16. The reported statistical significance between groups need to be reevaluated for all figures. Some statistically significant differences are irrelevant (e.g. Fig. 2D) and some statements in the manuscript are not reflected by statistically significant differences in the figures (e.g. Fig 2B). In Fig. 7, the way to show the results of statistical tests should be improved for easier data interpretation.

The presentation of Figure 7 has been changed substantially, including the separation of the data to individual graphs and showing significant differences of relevant comparisons. We believe that this enables a better interpretation and recognition of our conclusions. Statistical analyses have been re-evaluated resulting in individual adaptations of the respective test and changes in the depicted significant differences. Figure 2B is now Figure 2C. Knockdown of Chop widely blunted the induction of toxicity under NEN+Domp treatment conditions, indicating the Chop is an important mediator of the observed cytotoxic effects under combined treatment conditions. In contrast, ATF4 knockdown did not significantly reduced toxicity, suggesting that despite the induction of ATF4 upon combined treatment, this component of the ISR does not contribute to the downstream mechanisms resulting in cell death.

17. In Suppl. Fig. 1B, judged from the labeling at the bottom of the heatmap, the number of replicates does not match between groups. For example, in the NEN treatment group, where

are the replicates NEN_1 and NEN_4 (this reviewer wonders whether the researchers excluded certain replicates, but more importantly not even the same ones in each group)? The same applies to the "double treatment" group.

We agree that the description is unintentionally misleading. We isolated RNA from 5 replicates of each treatment, and selected 3 of each for the array analysis based on the quality and amount of the RNA of each isolate (therefore NEN 1 and 4 were deselected based on low RNA quality). For consistency we kept the original labeling "1-5" to be able to track the respective sample. The labeling was randomly chosen at the beginning, so that there is no connection between "Nr.1 of control" and "Nr. 1 of NEN" for example, which means there are no "same ones in each group".

18. The labeling "co" to refer to control conditions could be replaced by "ctrl" to adopt a more conventional nomenclature and avoid confusion when referring to the use of combined drugs.

This has been corrected for all Figures, thanks.

19. The graphical abstract in Figure EV5 and the manuscript mention "synergism", but this reviewer is not convinced that the researchers truly prove synergism with the performed assays.

Most (albeit not all) of the effects of combined drug treatments are more pronounced than the sum of single drug treatments, thereby meeting at least a trivial definition of synergism. Admittedly, we did not perform any specific and systematic analysis to prove this mathematically for the different readouts.

20. The title of Figure EV1 has a typo "ucoupling".

This has been corrected, thanks.

Referee #3 (Remarks for Author):

This study by Hartleben and colleagues, sets out on the premise (with good rationales) that combining metabolic inhibitors may make cancer cells more vulnerable to secondary therapies. On the back of this, they identify domperidone and various TCAs, as sensitisers to cell death in combo with mitochondrial uncoupler. Investigating the mechanism basis of this synergy they describe roles for autophagy, intergrated stress response, cholesterol metabolism converging on nucleotide metabolism to make cells more sensitive to DNA-damage (supported in the last figure). Study is interesting and timely, some aspects of the model proposed by the authors are supported by the corresponding data. However, in my

opinion, many of the findings here need further experimental support, to better demonstrate causality/elucidate mechanism. These points are detailed below:

We thank the reviewer for the positive general judgement that our study is interesting and timely. Please see below our responses to the specific points.

1. The demonstration that the ISR is being activated (and is relevant for cell death) needs to be better described, the data presented here are all consistent with activation of ER stress leading to apoptosis, given that CHOP but not ATF4 is relevant for the cell death observed upon combination treatment, ATF4 is also upregulated in the presence of ISRIB (Fig 2C), indicating that its upregulation is not dependent on ISR.

As acknowledged by the reviewer, we do see clear indication of activated integrated stress response (ISR), which we now also show with new data on P-eIF2a levels as well as ATF4 and CHOP protein expression data. ER-stress is part of the integrated stress response, but at this point, we do not know which specific arm of the ISR is activated. Interestingly, TFE3, as part of the ISR, can transcriptionally induce ATF4 (Martina et al., EMBO J (2016)35:479-495). Since ATF4 is still induced in the presence of ISRIB, we hypothesize that under the described conditions, ATF4 expression is largely driven transcriptionally via TFE3 and not translationally via P-eIF2a.

2. There is the implication in Fig 3. that in response to drug treatment TFE3, MITF are regulating autophagy and this is regulating metabolic homeostasis but this is never tested, i.e. does knockdown of either block the increase in autophagy observed and does inhibiting autophagy disrupt metabolic homeostasis (presumably increasing cell death) ?

Please note that Figure 3 has now been widely renewed based on the comments of reviewer #2 (point 6) demanding a more state-of-the-art assessment and interpretation of autophagy-related effects in the given context. We now show quantified immunofluorescence (IF) staining for Lamp 1, LC3-II foci and p62 (new Figure 3B), quantified immunoblots for LC3-II/I ratio and p62 (new Figures 3C and EV3C), as well as measurements of autophagic flux as the ratio of LC3-II protein levels in the absence or presence of chloroquine treatment under different drug treatment conditions (new Figure 3D). We confirmed the synergistic induction of autophagosome formation upon combined drug treatments (for single NEN, single Domp (IF), NEN+ Domp, NEN+Imiprimine (WB) or NEN+Amipramine (WB)), indicated by the marked induction of LC3-II (new Figures 3B, 3C and EV3C). This was in line with the induction of the CLEAR network genes. Interestingly, p62 levels showed a very similar pattern of regulation, indicating a blockage of autophagosome clearance at the level of lysosomal degradation, which was in agreement with the observed increase in the levels of the lysosomal marker Lamp1 (new Figures 3B, 3C and EV3C). Indeed, determination of autophagic flux by blocking lysosomal autophagosome

clearance using chloroquine confirmed that autophagic flux was reduced under combined drug treatment conditions despite induction of autophagosome formation (new Figure 3D). So, the interpretation has changed fundamentally from the previous assumption that the induction of autophagy might represent a response to induced metabolic stress (in an attempt to re-establish metabolic homeostasis) to the observed combined induction and blockage of autophagy presumably contributing to cell death under combined drug treatment conditions (in line with this reviewer's comment on a potential role of autophagy-blockage leading to cell death). It will be interesting to elucidate the exact molecular mechanism of autophagy blockage in the given context, which is however, beyond the scope of the present manuscript.

3. Gamma H2AX is used as a read out of DNA-damage that the authors conclude initiates cell death, however, DNA-damage occurs during cell death (as CAD cleaves DNA), whether the DNA-damage they observe is cause or consequence should be defined (for instance using caspase-inhibitors)

This is a valid point. We did the suggested experiment and see that rescue of cell death with the pan-caspase inhibitor zVAD also markedly reduced the levels of the DNA damage marker gamma-H2AX. Although we are not completely sure that the two processes can be clearly dissected in this manner, we acknowledge that this questions the previously proposed causality concerning DNA damage being a central mediator of the observed induction of cell death upon combined drug treatments. We have therefore modified the manuscript accordingly for not claiming direct causality. However, the observation that CHOP und UPP1 are central mediators of cell death in the given context is still valid, including a potential role of pyrimidine dyshomeostasis, as also suggested by the effects of DHODH inhibition upon sensitization with sublethal drug concentrations.

4. Given the translational lean of this paper (and of course of the journal), its important to investigate the potential toxicity of these drug treatments in normal cells/healthy tissue as far as is possible.

For addressing this point, we have now treated primary hepatocytes with all the individual drugs as well as the respective drug combinations using the concentrations and treatment duration usually used throughout the paper to induce cell death in cancer cells and determined caspase 3/7 activity (using Apo-ONE® Homogeneous Caspase-3/7 Assay; Promega) including staurosporin as positive control. As you can see in the new Figure 1F, primary hepatocytes did not show any considerable induction of caspase 3/7 activity levels upon treatment with the single and combined drugs. This is in line with our hypothesis that untransformed cells are less sensitive to the combined drug treatments, presumably through a better metabolic flexibility and less dependency on specific survival pathways in comparison to proliferative tumor cells.

5. Throughout the ms. cell viability/caspase activity is provided in relative levels, for key expts. its important to note the actual level (%) cell death, since, for instance, a relative increase of 3 fold caspase activity could equate to a 1 t 3% increase in cell death, clearly this would be be irrelevant when considering translating these findings.

We thank the reviewer for this very valid point. An additional readout confirming the relevance cell death effects upon combined drug treatments has also been requested by the other two reviewers (Reviewer #1, point 1 and Reveiwer #2 point 5). We therefore repeated the respective combined response (please see below).

As requested, we have now performed FACS-based analyses to asses cell death effects for single and combined drug treatments. We detected Annexin V- and PI-positive cells in order the determine the percentage of apoptotic cells along with staurosporin treatment as positive control as well as concomitant treatment with the pan-caspase inhibitor zVAD (for NEN+Domperidone combined treatments), the latter in order to counteract the induction of apoptosis under combined treatment conditions. The results are shown in the new Figures 1E (for HCT116 and U87 cells upon NEN/Domperidone single and combined treatments) and in Figures EV1E (for HCT116 and U87 cells for the combined treatments with NEN, Domperidone and the 4 tricyclic antidepressant (TCA) drugs).

The treatment conditions (drug concentrations and treatment durations) in the FACS analyses were identical to the conditions used for the respective toxicity or caspase assays shown previously. Under these conditions and depending on the cell line as well as the specific drug combinations, we observed up to 70%-80% apoptotic cells (combining Annexin V-positive cells in early apoptosis and AnnexinV/PI double-positive cells). In some approaches the proportion of apoptotic cells were lower (around 30%). However, please note that the analyses refer to fixed time point and the kinetics of cell death induction vary depending on the cell line as well as the different drug combinations. Upon treatment prologantion, combined treatments will result in all virtually complete death of cancer cells under investigation (in contrast to the treatment of untransformed cells (primary hepatocytes), now included in new Figure 1F in response to your point #4). The induction of apoptosis upon combined drug treatment were partly but significantly rescued by co-treating the cells with the caspase inhibitor zVAD. Additionally, we included samples treated in parallel with Staurosporin as a positive control for apoptosis induction. For each cell line, the length of the Staurosporin treatment (100µM) matched that of the other conditions (i.e 24 hours for the HCT116 cells and 48 hours for the U87) and resulted in comparable induction of cell death. We also tried respective analyses with the pancreatic cancer cell line BxPC3. However, the FACS-based analysis was substantially affected by the fact that these cells are very adherent and the required intense trypsinisation was not compatible with an appropriate conservation of cell integrity for the subsequent FACS analysis (which is generally more difficult with adherent cells). Taken together, the new analyses confirmed the relevance of cytotoxic effects of the combined drug treatments in HCT116 as well as U87 cells.

22nd Dec 2020

Dear Dr. Berriel Diaz,

Thank you for the submission of your revised manuscript to EMBO Molecular Medicine. We have now received the enclosed report from referee #3 who re-reviewed your manuscript. This referee also examined your answers to referee #1's report. Unfortunately, referee #2 could not return his/her report so far, and we therefore prefer to make a decision now in order not to delay the process further.

As you will see, referee #3 is now satisfied with your revised manuscript, and we will be able to accept your manuscript pending the following minor amendments:

1) Point-by-Point rebuttal letter: in your response to referee #2, point #13, you mention Figure EV7C. We can accommodate a maximum of 5 EV figures, and it is not clear to which figure you are referring to.

2) Main manuscript text

- Please answer/correct the changes suggested by our data editors in the main manuscript file (in track changes mode). This file will be sent to you in the next few days. Given the time of the year, please apologize a potential delay. When you receive this file, please use it for any further modification.

- Please remove the red text.

- Regarding the title of your manuscript, we think "Combination therapy induces lethal metabolic stress in cancer" is too vague, and would rather keep "Combination therapies induce cancer cell death through the (an?) integrated stress response and disturbed pyrimidine metabolism".

- Please provide up to 5 keywords.

- Please complete the Author contributions (Oliver Plettenburg is missing)

- Please add a "Conflict of interest" section (a title is missing)

- Please ensure that all funding is listed both in the Acknowledgements and in the submission system (they have to match).

- Material and methods:

- o Patients: Please confirm that the experiments conformed to the principles set out in the WMA Declaration of Helsinki and the Department of Health and Human Services Belmont Report (this should also be indicated in the checklist)

- o Cell lines: please indicate the origin and species of the cells used, as well as information on mycoplasma contamination and authentication (in the checklist as well).

- o Mice: please indicate the age of the mice used for hepatocyte isolation.

- o Antibodies: please indicate the dilutions that were used for immunoblot and immunofluorescence experiments

- o You sometimes refer to previously published methods. Please ensure that you nevertheless provide enough information so that the experiments are reproducible by the reader (in particular, please check the calculation of the Z' factor and the section on Patient-derived PDAC organoids and primary-dispersed cell lines

- Please indicate in the figures or in the legends the exact n= and exact p= values, not a range, along with the statistical test used. Some people found that to keep the figures clear, providing a supplemental table with all exact p-values was preferable. You are welcome to do this if you want to.

- Thank you for providing a Data Availability section. We note that the data is not yet public, please be aware that it should be done before acceptance of the manuscript. Please also provide the URL in the manuscript.

3) Figures:

- References to Fig 4 G,H ; Fig EV 4 B,C are missing. Please carefully check the callouts for all your figures in the main manuscript text.
- There are two tables in the material and methods: please make them Table 1 and Table 2 and add a reference in the text.
- Please add scale bars for figures 6A and EV3C. Please define the scale bar in Fig. EV1A

4) Thank you for providing Source Data. Please upload them so as to have 1 file per figure for the main figures, 1 file for all EV figures and 1 for all appendix figures.

5) Checklist: please complete the section E/12 and add a sentence stating that the experiments conformed to the principles set out in the WMA Declaration of Helsinki and the Department of Health and Human Services Belmont Report.

6) Thank you for providing the "Paper Explained" section. Could you please include it in the manuscript, and adopt the following structure: "Problem/results/impact". Please refer to any of our published articles for an example.

7) As part of the EMBO Publications transparent editorial process initiative (see our Editorial at <http://embomolmed.embopress.org/content/2/9/329>), EMBO Molecular Medicine will publish online a Review Process File (RPF) to accompany accepted manuscripts. In the event of acceptance, this file will be published in conjunction with your paper and will include the anonymous referee reports, your point-by-point response and all pertinent correspondence relating to the manuscript. Let us know whether you agree with the publication of the RPF and as here, **IF YOU WANT TO REMOVE OR NOT ANY FIGURES** prior to publication. Please note that the Authors checklist will be published at the end of the RPF.

I look forward to receiving your revised manuscript.

Yours sincerely,

Lise Roth

Lise Roth, PhD
Editor
EMBO Molecular Medicine

Photos 400-800 DPI

*Additional important information regarding figures and illustrations can be found at <https://bit.ly/EMBOPressFigurePreparationGuideline>

The system will prompt you to fill in your funding and payment information. This will allow Wiley to send you a quote for the article processing charge (APC) in case of acceptance. This quote takes into account any reduction or fee waivers that you may be eligible for. Authors do not need to pay any fees before their manuscript is accepted and transferred to our publisher.

***** Reviewer's comments *****

Referee #3 (Comments on Novelty/Model System for Author):

Authors have improved their ms. considerably during revision, all points I raised have been adequately addressed, either through experimentation or through textual modifications - the latter have not compromised the novelty or impact of the work.

Asked for my opinion on the authors' addressing Reviewer 1's comments I think they have addressed comprehensively all his/her comments

Referee #3 (Remarks for Author):

Authors have addressed all my comments comprehensively

The authors performed the requested editorial changes.

Corresponding Author Name: Mauricio Berriel Diaz

Journal Submitted to: EMBO Molecular medicine

Manuscript Number: EMM-2020-12461-V2